# Sexual dysfunction among Nigerian women living with HIV infection

Oliver Chukwujekwu Ezechi[1,2☯], Folahanmi Tomiwa Akinsolu[1,2☯]*, Tititola Abike Gbajabiamila[1☯], Ifeoma Eugenia Idigbe[1‡], Paschal Mbanefo Ezeobi[1,2☯], Adesola Zadiat Musa[1,2‡], Agatha Eileen Wapmuk[1☯]

1 Center for Reproduction and Population Health Studies, Clinical Sciences Department, Nigerian Institute of Medical Research, Yaba, Lagos State, Nigeria, 2 Department of Public Health, Faculty of Basic Medical and Health Sciences, Lead City University, Ibadan, Oyo State, Nigeria

☯ These authors contributed equally to this work.
‡ IEI and AZM also contributed equally to this work.
* folahanmi.tomiwa@gmail.com, Akinsolu.folahanmi@lcu.edu.ng

**Data Availability Statement:** All relevant data are within the manuscript.

**Funding:** The author(s) received no specific funding for this work.

## Abstract

### Introduction

Sexual dysfunction in women with HIV is a necessary but understudied aspect of HIV complications in women living with HIV. This study reports the prevalence, pattern, and risk factors for sexual dysfunction in women living with HIV in southwest Nigeria.

### Methods

A validated Female Sexual Function Index was used to determine sexual dysfunction in a cross-sectional study design involving 2926 adult women living with HIV in a large, publicly funded tertiary HIV treatment centre in Lagos, Nigeria. A score of less than 26.5 indicated sexual dysfunction. Multivariate logistic regression analysis was performed to identify risk factors for sexual dysfunction. P<0.05 was considered statistically significant at a 95% confidence interval (CI).

### Results

The prevalence of sexual dysfunction was 71.4%. The types of dysfunctions detected included disorder of desire (76.8%), sexual arousal (66.0%), orgasm (50.0%), pain (47.2%), lubrication (47.2%), and satisfaction (38.8%). Multivariate analysis showed that menopause (aOR: 2.0; 1.4–4.1), PHQ score of 10 and above (aOR: 2.3; 1.7–3.2), co-morbid medical conditions (aOR: 1.8; 1.4–2.7), use of protease inhibitor-based antiretroviral therapy (aOR: 1.3; 1.2–2.1) and non-disclosure of HIV status (aOR: 0.7; 0.6–0.8) were factors associated with sexual dysfunction.

### Conclusions

Sexual dysfunction is common among Nigerian women living with HIV. Menopause, use of protease inhibitor-based regimens, PHQ score of at least 10, co-morbid medical condition, and non-disclosure of HIV status were associated with sexual dysfunction. National HIV

**Competing interests:** The authors have declared that no competing interests exist.

programmes, in addition to incorporating screening and management of sexual dysfunction in the guidelines, should sensitise and train health workers on the detection and treatment of sexual dysfunction.

## Introduction

The introduction and increased access to antiretroviral therapy have changed human immunodeficiency virus (HIV) infection, a disease once considered a death sentence, to a chronic, manageable health condition. The widespread availability and use of potent combination antiretroviral therapy have dramatically reduced the morbidity and mortality of HIV infection globally and in sub-Saharan Africa [1–4].

The marked improvement in the clinical outcome of HIV infection has prompted a paradigm shift in the measure of HIV treatment success from mortality to quality of life [5]. Quality of life is a multidimensional construct defined as a subjective evaluation of one's functioning and well-being. It is affected by individual lifestyles and social norms, cultural practices, and belief systems, so it varies among regions and population groups [6, 7].

Numerous studies in sub-Saharan Africa and other parts of the world have examined the health-related quality of life in HIV-infected persons, with most studies emphasizing the physical, mental, and social aspects of quality of life while leaving out the sexual function component [6–21]. The few studies that addressed some aspect of sexual functioning were either limited by sample size or did not use a globally accepted and validated tool, such as the Female Sexual Function Index questionnaire, in conducting their study [11, 15, 17]. Few studies focused on sexual dysfunction among HIV-positive women, even though a high prevalence of sexual dysfunction in persons living with HIV/AIDS has been reported [22–24].

There are many reasons to expect an increase in sexual dysfunction in persons living with HIV infections, with psychological, endocrine, and neurologic factors being the most important [25–27]. However, proving a link between these factors and sexual dysfunction is difficult, as many of the conditions commonly seen within the HIV context, such as depression, peripheral neuropathy, and hypogonadism, are associated with sexual dysfunction in other settings [25, 27]. Furthermore, establishing associations with drugs or classes of drugs is difficult, as these agents often are used in combinations, and some of the drug-related toxicities persist even after replacement with another drug, leading to wrong attribution [25, 28, 29].

There is still a lack of research and management of related issues in African countries [30] despite the many studies on sexual dysfunction in persons living with HIV that have been conducted in developed countries [31, 32]. Also, several studies have evaluated sexual dysfunction in HIV-positive men [33], but few studies have examined the sexual function of women living with HIV/AIDS [25]. Despite the high burden of HIV in Nigeria, an extensive literature search has revealed only three studies reporting on sexual dysfunction among HIV-positive women [22–24], despite several studies on sexual dysfunction among women with chronic diseases, pregnancy, infertility, diabetes mellitus, or hypertension [34–39]. This study was conducted to determine the prevalence, pattern, and risk factors of sexual dysfunction among HIV-positive women in Lagos, Southwest Nigeria. This pivotal study has a threefold objective. First, we aim to determine the prevalence of sexual dysfunction among HIV-positive women in Lagos, offering crucial quantitative insights into how widespread this issue is within this demographic. Next, we delve into the pattern of sexual dysfunction. Here, "pattern" refers to the detailed breakdown of sexual dysfunction as explored through the six domains of the Female Sexual

Function Index (FSFI), which include desire, arousal, lubrication, orgasm, satisfaction, and pain. This comprehensive analysis allows us to understand not just the occurrence but the specific nature of sexual dysfunction experienced by these women. Lastly, we investigate the risk factors contributing to sexual dysfunction in this group, seeking to identify both the medical and psychosocial elements that might predispose these individuals to such challenges. By exploring these three critical areas, the study aims to provide a nuanced understanding of sexual dysfunction in HIV-positive women in Lagos, thereby laying the groundwork for more targeted and effective interventions.

## Methods

### Study setting, design, sample size, and sampling

This cross-sectional study was conducted between January 2019 and December 2020 at a large publicly funded tertiary comprehensive HIV treatment centre in a cosmopolitan city in southwestern Nigeria. The centre has cumulatively enrolled over 25,000 HIV-positive clients, 65% of whom are women. To participate in the study, the women had to be older than 18 years and proven HIV-positive for more than six months. The minimal sample size for the study was calculated with Raosoft online sample size calculator (http://www.raosoft.com/samplesize.html) [40], with a 99% CI, 2.5% margin of error, and assumed prevalence of sexual dysfunction among HIV positive to be 61.0% [11]. A total sample size of 2916 HIV-positive women was required, assuming a 30% non-response rate. A total of 3205 respondents randomly selected from the clinic attendance register were included in the study.

HIV-positive women attending the treatment centre were approached for the interview after selection from the lists of attendees, using a proportionate stratified sampling method. This method enabled the generalization of the study's findings to the women in the clinic. Each clinic's two-patient categories, drug refill; and 3–6 monthly physician consults, were considered a homogenous population. These two categories were often in the ratio of 2:1 at each of the three clinic days a week. The daily list of clinic attendees was categorized into the two groups above, each serving as a frame. Respondents were then selected from the frames by simple random sampling, using the ratio of 2:1. Those who accepted to participate and signed an informed consent form were recruited.

### Study tools, data collection, and statistical analysis

A self-administered structured data collection form consisting of three sections, (A) Demographic characteristics, (B) Patient Depression Question (PHQ), and (C) Female Sexual Function Index, were used to collect data for the study. The study tools were administered in English Language, and low-literate respondents were assisted in completing the forms by trained research assistants. The respondents' hospital numbers were written on each form to enable linkage to respondents' medical records. The demographic section of the form captured respondents' socioeconomic characteristics, medical and reproductive history, current and previous antiretroviral treatment, and recent laboratory results (CD4 cell count and viral load). The second section was the PHQ-9 [41, 42], a 9-item self-report questionnaire in which participants rated their feelings during the previous two weeks. Each question is scored 0 to 3 (0 = not at all, 1 = several days, 2 = more than half the days, and 3 = nearly every day), resulting in a 0 to 27. The nine items reflect the DSM-IV criteria for major depressive disorders. For this study, a score of 10 and above was considered depression. The last section is the Female Sexual Function Index (FSFI) [43, 44], a 19-item self-report instrument that scores six domains of sexual function in women in the previous four weeks. These domains include desire (two items, questions 1–2), arousal (four items, questions 3–6), lubrication (four items, questions

7–10), orgasm (three items, questions 11–13), satisfaction (three items, questions 14–16), and pain (three items, questions 17–19). The total FSFI score was determined by the sum of the six domains, with the minimum and maximum scores possibly varying from 2.0 to 36.0. A score greater than 26.5 denoted a low level of sexual dysfunction, and a score of 26.5 or less indicated sexual dysfunction. Likewise, scores on desire, arousal, lubrication, orgasm, satisfaction, and pain, which are ≤ 3.6 (score range, 1.2–6), ≤ 3.9 (score range, 0–6), ≤ 3.6 (score range, 0–6), ≤ 3.6 (score range, 0–6), ≤ 3.6 (score range, 0–6) and ≤ 4.4 (score range, 0–6), respectively; all indicate sexual dysfunction related to subdimensions [36, 43–45]. The FSFI has been validated on a clinically diagnosed sample of women with female sexual disorders [46]. In our setting [45], however, because the study was in a specific population, the validity and reliability in the study population were confirmed with 100 HIV-positive women. A Cronbach's alpha score of 0.85 was obtained, indicating good reliability.

The study results were presented as means and percentages. The total FSFI scores were categorized into dichotomous variables (normal sexual function and sexual dysfunction) using a cut-off point of 26.5 [44, 45]. Frequency distributions were generated, and univariate analysis using relevant statistics was performed to identify factors associated with sexual dysfunction. Multivariate logistic regression was used to identify independent risk factors for sexual dysfunction while controlling for potential confounders, including the severity of HIV disease measured by CD4 count and HIV viral load, time since HIV diagnosis, duration of antiretroviral drug use, PHQ-9 score of ≥ 10, co-morbid state, education, employment status, marital status, age, and menopausal state. Variables were entered into the model if their P value on univariate analysis was ≤ 0.25 or less. The variable with the strongest association in the univariate model was estimated first, followed by others in descending order. In the analysis, the comparison group consisted of those with an FSFI score of > 26.5 (no sexual dysfunction). P < 0.05 was considered statistically significant. Odds Ratios (OR) and 95% Confidence Intervals (CI) for the OR were also calculated. All analyses were performed with SPSS statistical software package version 23.0 for Windows, SPSS Inc., Chicago, IL, USA.

### Ethical considerations

Ethical approval for the study was obtained from the Institutional Review Board of the Nigerian Institute of Medical Research, Lagos, Nigeria. Written informed consent was obtained from participants after the study objectives and methods were explained to the participants. The participants were assured of the confidentiality and anonymity of their data during and after the study. The women who declined to consent to participate in the study were provided their routine care but excluded from participation in the study.

## Results

Out of 3205 HIV-positive women approached for participation in the study, a significant majority, 3009 (93.9%), consented and were either provided with the study forms or interviewed by research assistants. From this subset, 2926 (91.3%) women completed and returned the forms, thus providing a substantial and valuable dataset for our analysis.

### Sociodemographic and HIV-related characteristics of the respondents

The sociodemographic characteristics of the HIV-positive women in the study are shown in Table 1. Most of the women were less than 35 years of age (68.5%), married (53.1%), Christian (73.2%), working (59.5%), and from one of the three major Nigerian ethnic groups (75.8%). 78.5% had completed at least a secondary education. 2494 (85.2%) HIV-positive women were premenopausal.

**Table 1. Sociodemographic characteristics of 2926 HIV-positive women in the study.**

| Characteristics | No. of Respondents (%) |
|---|---|
| **Age (Years)** | |
| • < 35 | 2007(68.6) |
| • ≥ 35 | 919(31.4) |
| **Nigerian ethnic Groups** | 2218(75.8) |
| • Major ethnic groups (Igbo, H/Fulani, and Yoruba) | 708(24.2) |
| • Ethnic minorities | 629(21.5) |
| **Education** | 2296(78.5) |
| • < Secondary | 1741(59.5) |
| • ≥ Secondary | 1185(40.5) |
| **Employment status** | 1553(53.1) |
| • Working | 1372(46.9) |
| • Not working | 2142(73.2) |
| **Marital status** | 711(24.9) |
| • Married | 73(2.5) |
| • Not married | 2494(85.2) |
| **Religion** | 432(14.8) |
| • Christianity | |
| • Islam | |
| • Others | |
| **Menopausal state** | |
| • Premenopausal | |
| • Postmenopausal | |

HIV-related characteristics of the respondents are shown in Table 2, and the time elapsed since HIV diagnosis ranged from 6 to 204 months, with a median of 102 ± 41 months. In most respondents (56.5%), HIV diagnosis had been made within 120 months. Most of the women

**Table 2. HIV-related characteristics of the 2926 respondents in the study.**

| Characteristics | Frequency (%) |
|---|---|
| **Time since HIV diagnosis (months)** | |
| • < 120 | 1653(56.5) |
| • ≥ 120 | 1273(43.5) |
| **CD4 counts (2867)** | 757(26.4) |
| • < 200 | 2110(73.6) |
| • ≥ 200 | 563 (194) |
| • Median (SD) | 1569(62.3) |
| **Viral load (2519)** | 950(37.7) |
| • Not detected | 1065(43.5) |
| • Detected | 1383(56.5) |
| **Partner HIV status (2448)** | 1807(66.3) |
| • Negative | 918(33.4) |
| • Positive | 2051(70.1) |
| **HIV status Disclosure (2725)** | 874(29.9) |
| • Disclosed | 1743(59.6) |
| • Not disclosed | 1182(40.4) |
| **Type of antiretroviral therapy regimen** | 101(31) |
| • NNRTI based | 509(20.2) |
| • PI based | 2016(79.8) |
| **Duration on ART (months)** | 564(21.9) |
| • < 96 | 2011(78.1) |
| • ≥ 96 | |
| • Median (SD) | |
| **Opportunistic infection (2525)** | |
| • Yes | |
| • No | |
| **Co-morbid medical condition (2575)** | |
| • Yes | |
| • No | |

**Table 3. Distribution of 2926 respondents PHQ-9 total score.**

| Total score | Number of Respondents (%) |
|---|---|
| 5–9 | 2495(85.3) |
| 10–14 | 313(10.7) |
| 15–19 | 93(3.2) |
| $\geq 20$ | 25(0.8) |
| Range | 0–25 |
| Mean (SD) | 5.6 (2.3) |

had CD4 cell counts above 200 cells/mm$^3$ (73.6%), not detectable HIV viral load (62.3%), HIV positive partners (56.5%), and had disclosed their HIV status to their partner (66.3%). Of the 2926 respondents in the study, 2051 (70.1%) received a non-nucleoside reverse transcriptase inhibitor-based regimen, and 874 (21.9%) received a protease inhibitor-based regimen. The duration of antiretroviral therapy (ART) among the respondents ranged from 6 to 204 months, with a median of 101± 31 months. Whereas 1743 (59.6%) of the respondents treated with ART have been receiving it for less than 96 months, the remaining 1182 (52.6%) had been receiving ART for 95 months or more. At the time of the interview, 509 (20.2%) women had opportunistic infections, and 564 (21.9%) had one or more co-morbid medical conditions.

## Respondents' PHQ-9 total score

The total score ranged from 0–25, with a mean of 5.6 ± 2.3. Most respondents had PHQ-9 scores of less than 10 (85.3%), and the remaining respondents (14.7%) had PHQ-9 scores of 10 and above. Only twenty-five respondents (0.8%) had a total score of 20 and above (See Table 3).

## Sexual dysfunction among the respondents

Of the 2926 women with complete questionnaires, 2089 had FSFI score $\leq$ 26.6 or a sexual dysfunction rate of 71.4% (CI: 68.4–75.3). The distribution of respondents with sexual dysfunction in each of the six domains of sexual function is shown in Table 4. The percentage of sexual dysfunction in the domains ranged from 38.8% in the satisfaction domain to 76.8% in the desire domain. Over half of the respondents in this study reported experiencing dysfunction in several key sexual function domains: specifically, 76.8% experienced issues with desire, 66.0% with arousal, and 50.0% faced challenges with achieving orgasm. The proportion of respondents with dysfunction in pain, lubrication, and satisfaction domains was 47.2%, 47.2%, and 38.8%, respectively.

## Risk factors for sexual dysfunction among the cohort

Table 5 shows the relationship between the women's sociodemographic characteristics, co-morbid conditions, HIV characteristics, and sexual dysfunction. On univariate analysis, a significant association was found between women age $\geq$ 35 years (cOR: 1.4; 1.0–1.8), marital

**Table 4. Distribution of domain-specific sexual dysfunction among the 2926 respondents in the study.**

| Sexual Domains | Number of respondents with sexual dysfunction (%) |
|---|---|
| Desire | 2247 (76.8) |
| Arousal | 1931 (66.0) |
| Lubrication | 1381 (47.2) |
| Orgasm | 1463 (50.0) |
| Satisfaction | 1135 (38.8) |
| Pain | 1381 (47.2) |

**Table 5. Relationship between sociodemographic variables and sexual dysfunction among 2926 HIV-positive women in the study.**

| Variables | Sexual dysfunction N = 2089 | Normal sexual function N = 837 | Crude Odds Ratio (95% CI) | P value | Adjusted Odds Ratio (95% CI) | P value |
|---|---|---|---|---|---|---|
| **Age(years)** | | | | | | |
| • <35 | 1393(69.4) | 614(30.6) | 1.0 | 0.04 | 1.0 | 0.09 |
| • ≥ 35 | 696(75.7) | 223(24.3.) | **1.4(1.0–1.8)** | 0.51 | 1.1(0.8–1.5) | - |
| **Education** | 447(71.1) | 182(28.9) | 0.9(0.7–1.14 | 0.31 | - | - |
| • < Secondary | 1642(71.4) | 655(28.6) | 1.0 | 0.00 | - | 0.05 |
| • ≥ Secondary | 1253(72.0) | 488(28.0) | 1.1(0.8–1.4) | 0.47 | - | - |
| **Employment status** | 836(70.5) | 349(29.5) | 1.0 | **0.000** | - | **0.000** |
| • Working | 1062(68.4) | 491(31.6) | **0.5(0.4–0.7)** | **0.013** | 0.9(0.6–1.1) | 0.02 |
| • Not working | 1027(74.8) | 346(25.2) | 1.0 | **0.02** | 1.0 | **0.001** |
| **Marital status** | 1525(71.2) | 617(28.8) | 1.0(0.7–1.3) | | - | |
| • Married | 504(70.9) | 207(29.1) | 1.0 | | - | |
| • Not married | 1723(69.1) | 772(30.9) | 1.0 | | 1.0 | |
| **Religion** | 366(84.9) | 65(15.1) | **2.5(1.6–4.0)** | | **2.3(1.7–3.2)** | |
| • Christianity | 370(85.7) | 62(14.3) | **2.5(1.1–5.6)** | | **2.0(1.4–4.1)** | |
| • Islam | 456(80.8) | 108(19.2) | **1.8(1.1–3.0)** | | **1.8(1.4–2.7)** | |
| **PHQ Score** | | | | | | |
| • <10 | | | | | | |
| • ≥10 | | | | | | |
| **Postmenopausal status** | | | | | | |
| **Co-morbid medical disorders** | | | | | | |

NNRTI: Nucleoside reverse transcriptase inhibitors; PI: Protease inhibitor

status (cOR:0.5;0.4–0.7), PHQ Score of ≥10 (cOR: 2.5; 1.6–4.0), postmenopausal status (cOR: 2.5; 1.1–5.6), co-morbid medical disorders (cOR: 1.8; 1.1–3.0) and sexual dysfunction. No statistically significant association was found between education (p = 0.51), religion (p = 0.47), employment status (p = 0.31), and sexual dysfunction.

The relationship between HIV-related factors and sexual dysfunction is summarized in Table 6. A statistically significant association was also found between HIV status disclosure

**Table 6. Relationship between HIV-related variables and sexual dysfunction among 2926 HIV-positive women in the study.**

| HIV Variables | Sexual dysfunction N = 2089 | Normal sexual function N = 837 | Crude OR (95% CI) | P value | Adjusted OR (95% CI) | P value |
|---|---|---|---|---|---|---|
| **Time since HIV diagnosis (months)** | | | | | | |
| • < 120 | 1160(70.2) | 493(29.8) | 0.9(0.7–1.1) | 0.18 | - | - |
| • ≥ 120 | 929(72.9) | 344(27.1) | 1.0 | 0.81 | - | - |
| **CD4 count** | 560(74.0) | 197(26.0) | 1.8(0.6–3.1) | 0.1 | - | - |
| • <200 | 1529(72.5) | 581(27.5) | 1.0 | 0.60 | - | - |
| • ≥ 200 | 1092(69.6) | 477(30.4) | 0.8(0.6–1.0) | 0.04 | - | **0.03** |
| **Viral load** | 590(62.1) | 360(37.9) | 1.0 | 0.01 | - | **0.01** |
| • Not detected | 1001(72.4) | 383(27.6) | 1.1(0.8–1.5) | 0.00 | - | 0.06 |
| • Detected | 753(70.7) | 312(29.4) | 1.0 | 0.01 | - | 0.07 |
| **Partner HIV status** | 1250(69.2) | 557(30.8) | **0.7(0.6–0.97)** | | **0.7(0.6–0.8)** | |
| • Positive | 692(75.4) | 226(24.2) | **1.0** | | **1.0** | |
| • Negative | 135866.2) | 693(33.9) | **1.0** | | **1.0** | |
| **HIV status disclosure** | 664(76.0) | 210(24.0) | **1.6(1.1–2.4)** | | **1.3(1.2–2.1)** | |
| • Disclosed | 1333(76.5) | 410(23.5) | **1.5(1.1–2.1)** | | 1.4(0.9–2.4) | |
| • Not disclosed | 801(67.8) | 381(32.2) | **1.0** | | 1.0 | |
| **Antiretroviral drug regimen** | 401(78.8) | 108(21.2) | **1.6(1.1–2.4)** | | 1.4(0.8–1.7) | |
| • NNRTI based | | | | | | |
| • Protease Inhibitor-Based | | | | | | |
| **Duration on HAART (month)** | | | | | | |
| • < 96 | | | | | | |
| • ≥ 96 | | | | | | |
| **Had opportunistic infection** | | | | | | |

(cOR:0.7; 0.6–0.97), use of protease inhibitor-based antiretroviral drug regimen (cOR: 1.6; 1.1–2.4), duration of HAART <96 months (cOR:1.5; 1.1–2.1), the presence of opportunistic infection (cOR: 1.6(1.1–2.4) and sexual dysfunction. Time since HIV diagnosis (p = 0.18), CD4 count (p = 0.8), Viral load (p = 0.1), and partners' positive HIV status (p = 0.6) have no significant association with sexual dysfunction.

After adjustment for potential confounding variables of age, PHQ score, time since HIV diagnosis, HIV status disclosure, antiretroviral drug regimen, presence of opportunistic infection, and co-morbid medical conditions in the multivariate logistic regression model, sexual dysfunction in HIV-positive women had an independent statistically significant association with the postmenopausal state (aOR: 2.0; 1.4–4.1), PHQ score of ≥10 (aOR: 2.3; 1.7–3.2), co-morbid medical disorders (aOR: 1.8; 1.4–2.7), use of protease inhibitor-based regimen (aOR: 1.3; 1.2–2.1), and HIV status disclosure (aOR: 0.7; 0.6–0.8).

## Discussion

Whereas several studies reporting the prevalence and risk factors for sexual dysfunction in the general population and select medical conditions abound [41, 43–46], data on sexual dysfunction among HIV-positive women is limited [11]. In Nigeria, with the second largest HIV burden globally, the situation is more dire, as only three publications have reported on sexual dysfunction among HIV-positive women [22, 23, 25]. The three Nigerian studies were conducted in a predominantly Muslim population with a relatively small sample size, making the findings difficult to generalize. In this study, we determined the prevalence, pattern, and risk factors for sexual dysfunction using information obtained from a large population of HIV-positive women residing in a cosmopolitan city that reflected sociocultural diversity who had been receiving antiretroviral therapy for much longer. The larger sample size also made it possible to conduct subgroup analyses that are powered to make generalization easier. Information obtained in the study will assist in identifying HIV-positive women with sexual dysfunction within the country, within our subregion, and elsewhere to provide adequate care.

The sexual dysfunction rate of 71.4% in this study is within 61.0–89.2% reported in Nigerian studies [23–25]. It is, however, higher than the rates reported among HIV-positive women in India [47] and Europe [48, 49]. Although female sexual dysfunction is a widespread health condition, controversy exists regarding the burden of female sexual dysfunction; vast differences are reported within and between countries. The differences may reflect medical and psychological factors, especially in socioeconomic, cultural, and ethnic differences, the definition used for each type of dysfunction, the type of study tool, the method of data collection, samples, and characteristics of populations studied [22, 23, 25, 47–51]. In addition, the observed higher prevalence in our setting may be due to cultural differences in the settings or the reluctance of women in the sub-Saharan region to talk openly about sex and seek solutions for their sexual concerns [11, 22, 23, 25, 47–51]. The various explanations for the observed differences have also made generalizations and comparisons of studies challenging. This study highlights the significant burden of sexual dysfunction within the HIV-positive population in Nigeria. This concern is particularly acute when considered against the backdrop of the already high prevalence (63%) of sexual dysfunction among women in general in the country [52]. This study used a validated and standardized tool and a specific population group to ensure generalization to our population group and context. This study confirmed that HIV infection is associated with high sexual dysfunction and thus deserves close attention [22, 23, 25, 47–51].

The result PHQ-9 score of ≥10 is (cOR) of 2.5 (95% CI: 1.6–4.0) shows that sexual dysfunction is associated with depressive symptoms. A study by Atlantis & Sullivan (2012) found a

strong link between depression and sexual dysfunction, suggesting that the presence of depressive symptoms can significantly increase the likelihood of experiencing difficulties with sexual function [53]. A study by Clayton et al. (2014) found similar results, noting that depression was a significant predictor of sexual dysfunction in women, and suggested that treatment of depressive symptoms could have beneficial effects on sexual function [54]. The study by Mathew & Weinman (2009) also found a strong association between depression and sexual dysfunction, further supporting the findings of this study [55]. However, while these results are compelling, it's essential to consider them in the context of potential confounders and the complexity of sexual health. Factors such as medication side effects, relationship issues, and other psychological or physical health conditions can also significantly impact sexual function. The relationship between depression and sexual dysfunction is likely to be bidirectional and influenced by a range of other factors [56, 57].

The present study found that the postmenopausal state (aOR: 2.5), PHQ score of at least 10 (aOR: 2.5), opportunistic infection (aOR:1.4), co-morbid medical condition (aOR: 1.8), use of protease-based antiretroviral regimen (aOR: 1.3), and HIV status disclosure (aOR: 0.7) were independently associated with sexual dysfunction in women living with HIV infection. The association between changes in blood hormonal levels in women during menopause and a decline in sexual functioning is well-established in the scientific literature. Studies have consistently demonstrated that the hormonal imbalances experienced during menopause, particularly the reduction in estrogen and testosterone levels, are significantly linked to various aspects of sexual dysfunction. In a comprehensive review by Nappi and Lachowsky (2009) highlights the direct correlation between decreased estrogen levels and reduced vaginal lubrication, decreased libido, and other sexual health issues in menopausal women [58]. Menopausal status was also found as a risk factor for sexual dysfunction in other studies involving women with known HIV status and the general population [11, 59–61]. These changes occur in all women, not only those HIV-positive. The presence of opportunistic infection and comorbid medical conditions, such as hypertension and diabetes, were also identified as risk factors for sexual dysfunction among women in this study. A common opportunistic infection among the cohort in our clinic is tuberculosis [62]. Available literature has reported that active tuberculosis disrupts sexual function in male and female patients. Tuberculosis usually disturbs the mental and physical well-being of those infected. Its chronic nature, long isolation period, and enormous pill burden taken over a long period may lead to sexual dysfunction. Tuberculosis of the female genital tract is one of the leading causes of reproductive dysfunction as well [63].

Sexual dysfunction is frequently a challenge in patients with other medical diseases, such as hypertension and diabetes, and it may occur either as a side effect of medications or as a component of the dysmetabolic syndrome of high blood pressure [33, 39]. Hypertension and diabetes are prevalent comorbidities in our environment, particularly among the HIV patient population [64], and 61–88% of patients with these diseases experience sexual dysfunction [11, 25, 37, 65]. While erectile dysfunction is a well-recognized consequence of hypertension in men, women with hypertension also face significant sexual health challenges, including reduced sexual desire, arousal issues, and overall dissatisfaction with sexual life. The conclusive evidence of this association between hypertension and various forms of sexual dysfunction in women is well-documented in a systematic review and meta-analysis by Santana, L.M., Perin, L., Lunelli, R. et al. (2019) [66]. Given the high prevalence of sexual dysfunction among patients with these comorbid conditions, it is not surprising that co-morbid medical conditions retained an independent statistical association with sexual dysfunction, even after controlling for other possible confounders. This association is consistent with findings from other studies in our environment [11, 35, 37, 66]. The use of antiretroviral therapy has been shown

to ameliorate sexual dysfunction in several studies [11], however, we could not access this variable in our study, as it is now unethical not to start HIV-positive patients on antiretroviral therapy; the studies that reported the association between sexual dysfunction and the use of antiretroviral therapy were conducted before the treat-all policy was instituted. In our study, we aimed to evaluate the potential association between the type of antiretroviral therapy regimen and sexual dysfunction in HIV-positive women. Consistent with findings from previous studies [67, 68]. In a study conducted by by Schrooten et al. in 10 European countries, we observed that the use of protease inhibitors was an independent risk factor for sexual dysfunction [65]. Specifically, Schrooten et al. reported a notable decrease in sexual interest and potency among HIV-positive patients receiving protease inhibitors compared to those who were protease inhibitor-naïve [65].

As anticipated, the occurrence of abnormalities in specific sexual function domains is more pronounced in the Nigerian HIV-positive population studied than in the general Nigerian population. However, this frequency aligns with what has been reported in other studies focusing on HIV-positive women, indicating a consistent pattern of sexual dysfunction among HIV-positive women across different geographical and cultural contexts [25, 48–50, 69]. Previous studies reported an association between HIV infection and poor quality of life resulting from physical, psychological, or emotional factors [6–8, 13, 18]. Medical conditions, including HIV infection, reduce the quality of life and thus impact sexual function [11, 15]. Findings in this study support this assertion, as a co-morbid medical condition (aOR: 1.8; 1.1–3.0) and patients' health questionnaire scores of at least 10 (aOR: 2.3; 1.7–3.2) retained independent statistically significant association with sexual dysfunction after controlling for potential confounding variables. In this study, a significant majority of respondents experienced dysfunction in at least three specific sexual function domains, with 76.8% reporting issues with desire, 66.0% with arousal, and 50.0% with orgasm. A large proportion also had dysfunction in the remaining three sexual function domains: pain (47.2%), lubrication (47.2%), and satisfaction (38.8%) domain. These findings are similar to previous studies conducted in Nigeria and other African countries among HIV-positive women, in which the most prevalent type of sexual dysfunction was in the domain of desire.

In contrast, the least was in the domain of pain [11, 22–25, 69]. The above finding, however, is higher than the findings of studies among the general population. Still, the distribution of degrees of dysfunction in the various domains was like the findings in women living with HIV [35, 61, 70–73]. Across most studies, dysfunction in the desire domain was the most common, and dysfunction in the pain domain was the least common, as found in this study. Castelo-Branco et al. [71], in their study among healthy women living in Santiago de Chile, found desire disorder (38.0%) the most common, followed by arousal disorder (32.0%), pain disorder (33%), and orgasm disorder (25.0%). In a study by Kadri and colleagues [73] among women living in Morocco, desire disorder was reported by 18%, orgasm disorder by 12%, arousal disorder by 8%, and pain disorder by 8%. The inference that can be made from the above is that while the severity of specific domain dysfunction may be worse in HIV-positive women, their distribution and pattern of occurrence are like those in HIV-positive and HIV-negative women [11, 22, 23, 25, 61, 73–76].

Castelo-Branco et al. [71], in their study among healthy women living in Santiago de Chile, found desire disorder (38.0%) the most common, followed by arousal disorder (32.0%), pain disorder (33%), and orgasm disorder (25.0%). In a study by Kadri and colleagues [73] among women living in Morocco, desire disorder was reported by 18%, orgasm disorder by 12%, arousal disorder by 8%, and pain disorder by 8%. The inference that can be made from the above is that while the severity of specific domain dysfunction may be worse in HIV-positive

women, their distribution and pattern of occurrence are like those in HIV-positive and HIV-negative women [11, 22, 23, 25, 61, 73–76].

## Strengths and limitations

Our study has strengths: It was conducted in an HIV-endemic country with a large sample size using an internationally accepted and validated tool. The study was also conducted in a cosmopolitan city with sociocultural, religious, and economic diversity mimicking what is present nationally. However, the study also has limitations: its cross-sectional design did not allow for the assessment of the temporal association between sexual dysfunction and various factors studied. The study was conducted in a single city and clinic, though of more than 20 million population and over 20,000 patients; this makes it difficult to generalize to all women living with HIV in Nigeria. FSFI, the study tool, has been criticized for considering only penetrative sex as the single requirement for sexual well-being. Finally, about 10% of eligible patients who were approached declined to participate, failed to return the data collection form, or returned incompletely answered forms;. However, the rate was relatively low, it could still have affected the generalizability of the study results.

## Conclusions

Sexual dysfunction is a significant public health challenge among HIV-positive women in Nigeria, where HIV is endemic. The sexual dysfunction rate of 71.4% confirms that it is a significant challenge in our setting. Sexual dysfunction in the six domains ranged from 38.8% in the satisfaction domain to 76.8% in the desire domain. Six factors are significantly associated with sexual dysfunction: menopausal state, PHQ score of 10 and above, opportunistic infection, co-morbid medical condition, use of protease-based antiretroviral regimen, and non-disclosure of HIV status. There is a need to sensitize and train healthcare workers to assess for sexual challenges during routine programmatic consultations. The national HIV programs should, as a matter of urgency, include screening and treatment of sexual dysfunctions among HIV-positive women in the country's treatment guidelines.

## Author Contributions

**Conceptualization:** Oliver Chukwujekwu Ezechi, Paschal Mbanefo Ezeobi, Agatha Eileen Wapmuk.

**Data curation:** Oliver Chukwujekwu Ezechi, Folahanmi Tomiwa Akinsolu, Tititola Abike Gbajabiamila, Ifeoma Eugenia Idigbe, Paschal Mbanefo Ezeobi, Adesola Zadiat Musa, Agatha Eileen Wapmuk.

**Formal analysis:** Oliver Chukwujekwu Ezechi, Tititola Abike Gbajabiamila, Ifeoma Eugenia Idigbe, Paschal Mbanefo Ezeobi, Adesola Zadiat Musa.

**Investigation:** Tititola Abike Gbajabiamila, Ifeoma Eugenia Idigbe, Adesola Zadiat Musa.

**Methodology:** Oliver Chukwujekwu Ezechi, Folahanmi Tomiwa Akinsolu, Tititola Abike Gbajabiamila, Adesola Zadiat Musa.

**Resources:** Folahanmi Tomiwa Akinsolu.

**Software:** Paschal Mbanefo Ezeobi.

**Supervision:** Oliver Chukwujekwu Ezechi, Folahanmi Tomiwa Akinsolu, Ifeoma Eugenia Idigbe, Paschal Mbanefo Ezeobi.

**Validation:** Folahanmi Tomiwa Akinsolu, Adesola Zadiat Musa.

**Visualization:** Adesola Zadiat Musa.

**Writing – original draft:** Oliver Chukwujekwu Ezechi, Folahanmi Tomiwa Akinsolu, Tititola Abike Gbajabiamila, Paschal Mbanefo Ezeobi, Agatha Eileen Wapmuk.

**Writing – review & editing:** Oliver Chukwujekwu Ezechi, Folahanmi Tomiwa Akinsolu, Tititola Abike Gbajabiamila, Paschal Mbanefo Ezeobi, Agatha Eileen Wapmuk.

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
