## [Decision Letter · Decision Letter 0]

6 Dec 2023

PONE-D-23-24987Sexual Dysfunction among Nigerian Women Living with HIV InfectionPLOS ONE

Dear Dr. Akinsolu,

Thank you for submitting your manuscript to PLOS ONE. After careful consideration, we feel that it has merit but does not fully meet PLOS ONE’s publication criteria as it currently stands. Therefore, we invite you to submit a revised version of the manuscript that addresses the points raised during the review process.

We look forward to receiving your revised manuscript.

Kind regards,

Deidre Pretorius, PhD

Academic Editor

PLOS ONE

Journal Requirements:

Additional Editor Comments (if provided):

ongratulations with research well conducted this body of work will make a good contribution to the field of female sexual dysfunction (FSD).

Please consider the following comments:

Who is the corresponding author? An * is missing at one of the names.

Introduction:

Line 52: Review the need of including reference 1-3? Throughout the text, the first 3 references are never used. Consider removal and renumbering.

Line 58: “Numerous studies in sub-Saharan Africa” Reference 7 refers to a study done ins Tehran. Kindly motivate the use when referring to Sub-Saharan Africa.

Line 60: Reference 7-21, should if not be 8-19?

Line 72: “But few studies”. Consider the grammar: but a few or only a few.

Line 76: Relevance of ref 31-36: do you need all this references? List is quite long.

Reference 31& 33: these studies are about Hypertension and ED. Please motivate the use in a study on FSD.

After reading the article, I suggest that you review the introduction. It needs to be more background information.

If sexologists read this work, they would be interested in the prevalence of sexual dysfunction in the general population in Nigeria. Mention the prevalence amongst the general population, followed by the prevalence in men, then in women. The aim of the study is to establish the prevalence in the female HIV+ population.

Address HIV burden in Nigeria – set the scene to the reader will give great context.

Include a short sentence of HIV treatment regimens offered in Nigeria.

Provide Examples of opportunistic infection and co-morbid medical conditions.

Line 76: Conclude the Introduction section with the aim of the study clearly explained.

“The study was conducted to determine the

• Prevalence

• Pattern

• Risk factors of sexual dysfunction in HIV + Women in Lagos.

“Pattern” – elaborate on what you mean: The 6 domains tested with FSFI?

Methods

Line 98: Consider the word included instead of recruited.

Line 100: What language version was used in both these measures? If English, how did you ascertain language proficiency in order to self-report?

Line 102: It is not clear why you decided to combine the PHQ with the FSFI?

Depression is a known risk factor for sexual dysfunction in all age groups, gender, and populations. There is no reference depression in the introduction and it kind of surprises the reader that it was used. Include your reasoning on using the PHQ in the introduction.

Line 121: The information about validity and reliability assessed in 100 pts prior to conducting the study does not constitute a conventional psychometric evaluation process. How was validity (in the population under investigation) of the instruments determined? This could be viewed as a limitation.

Results

Line 150: “women in a stable monogamous and heterosexual relationship.” On what grounds are you able to make this statement? These determinants were not part of your sociodemographic information collected. How do you know this? Revision of this strong statement recommended.

Line 156: Why not report the 85.2% that are premenopausal? All the other sociodemographic characteristics, you report the higher %? Consistence is important.

Line 160: Are all partners tested in Nigeria? The options are only positive or negative partners. Are there no partners with their status unknown? Maybe alluded to this in the Introduction?

Line 164: Consistency recommended: (1743,59.6%) of the respondents. Rather 1743 (59.6%) of the respondents.

Line 169-172: Mention table 3 (consistency)

Line 177: “At least half of the respondents” = with scores of 76,66 and 50%? Review the sentence since it is way more than half?

Line 185: SD – it is the first time you use this abbreviation. What does it stand for? Standard deviation or sexual dysfunction? If it is sexual dysfunction, insert (SD) after the words “sexual dysfunction” in line 182. Or where you use it the first time in the article.

Page 20: Table 5 and 6 should be split in 2, each with its separate heading.

Line 194 -198: Consider mentioning the multivariates in the order as mentioned on table 5 &6:

Age, marital status, PHQ score >10, postmenopausal state, co-morbid medical conditions, HIV disclosure status, use of PI’s, duration of ART < 96 months, presence of opportunistic infection.

Highlight the factors in the same order for the sake of consistency: PHQ score >10, postmenopausal state, co-morbid medical conditions, HIV disclosure status, use of PI’s.

Discussion:

Line 220-221: “our setting and Sub-Saharan Africa” this is confusing. The study was only conducted in a single city in Nigeria as mentioned in line 207 – 210. Consider the removal of “and Sub-Saharan Africa”.

Line 225 -226: it will have more impact if you refer to the prevalence of Sexual dysfunction in women in general in Nigeria and highlight the high burden of sexual dysfunction in the HIV + population.

Line 228: is this referring to global frequency of abnormality or to the Nigerian setting?

Line 235: “More than half of the respondents” – review sentence as advised before.

Line 254: Menopausal status instead of “A menopausal state” maybe better to use.

Line 256: it is Possible – why not quote a study that prove the link with a hormonal imbalance in menopause with sexual dysfunction? To say it is possible, leave it open for speculation and I think science has established this association long ago.

Line 267 & 271: Ref 33 – Erectile dysfunction is associated with Hypertension and this study is done on women. Although it is a well-known sexual problem associated with hypertension, how relevant is this reference in an article reporting on findings in a female population? Do we have and specific sexual dysfunction associated in females with hypertension in the literature? Maybe refer to this systematic review for further research?

Santana, L.M., Perin, L., Lunelli, R. et al. Sexual Dysfunction in Women with Hypertension: a Systematic Review and Meta-analysis. Curr Hypertens Rep 21, 25 (2019). https://doi.org/10.1007/s11906-019-0925-z

Line 280: the word potency – what does it mean? Sexual interest and potency? Why not use one of the 6 domain words from the FSFI like desire?

Line 284-289: Discussion disclosure of HIV status and the fears surrounding it and referring the violence – please consider reference this statement?

The discussion address 5 of the 6 independently associated risk factors identified. There is no discussion about the PHQ score >10. As in the introduction – it is really not making sense why this has been included in the study methodology and mentioned in the results, but not put into perspective in the introduction and not discussed in the discussion? (It should be included, the evaluation of Depression is relevant to FSD). Address the omission in the 2 sections: Introduction and Discussion.

Consider consistency. Discuss the 6 factors in the exact order as before:

• PHQ score >10,

• postmenopausal state,

• co-morbid medical conditions,

• HIV disclosure status,

• use of PI’s.

Keeping the same order makes the reading and understanding of the findings so much easier to read and remember for a reader, especially if you have a non-academic reading the paper.

Conclusion

Line 304 – 305: why include Sub-Saharan Africa in this sentence?

Reviewers' comments:

Reviewer's Responses to Questions

**Comments to the Author**

1. Is the manuscript technically sound, and do the data support the conclusions?

Reviewer #1: Yes

Reviewer #2: Yes

2. Has the statistical analysis been performed appropriately and rigorously? 

Reviewer #1: Yes

Reviewer #2: Yes

3. Have the authors made all data underlying the findings in their manuscript fully available?

Reviewer #1: Yes

Reviewer #2: Yes

4. Is the manuscript presented in an intelligible fashion and written in standard English?

Reviewer #1: Yes

Reviewer #2: Yes

5. Review Comments to the Author

Reviewer #1: well conducted and important study for PLWHIV and health professionals .involved in HIV care While the incidence figures are higher than in other studies it would be interesting to the incidence data for mixed sexual dysfunctions or patients with more than one dysfunction like desire and orgasm etc

Reviewer #2: Congratulations with research well conducted this body of work will make a good contribution to the field of female sexual dysfunction (FSD).

Please consider the following comments:

Who is the corresponding author? An * is missing at one of the names.

Introduction:

Line 52: Review the need of including reference 1-3? Throughout the text, the first 3 references are never used. Consider removal and renumbering.

Line 58: “Numerous studies in sub-Saharan Africa” Reference 7 refers to a study done ins Tehran. Kindly motivate the use when referring to Sub-Saharan Africa.

Line 60: Reference 7-21, should if not be 8-19?

Line 72: “But few studies”. Consider the grammar: but a few or only a few.

Line 76: Relevance of ref 31-36: do you need all this references? List is quite long.

Reference 31& 33: these studies are about Hypertension and ED. Please motivate the use in a study on FSD.

After reading the article, I suggest that you review the introduction. It needs to be more background information.

If sexologists read this work, they would be interested in the prevalence of sexual dysfunction in the general population in Nigeria. Mention the prevalence amongst the general population, followed by the prevalence in men, then in women. The aim of the study is to establish the prevalence in the female HIV+ population.

Address HIV burden in Nigeria – set the scene to the reader will give great context.

Include a short sentence of HIV treatment regimens offered in Nigeria.

Provide Examples of opportunistic infection and co-morbid medical conditions.

Line 76: Conclude the Introduction section with the aim of the study clearly explained.

“The study was conducted to determine the

• Prevalence

• Pattern

• Risk factors of sexual dysfunction in HIV + Women in Lagos.

“Pattern” – elaborate on what you mean: The 6 domains tested with FSFI?

Methods

Line 98: Consider the word included instead of recruited.

Line 100: What language version was used in both these measures? If English, how did you ascertain language proficiency in order to self-report?

Line 102: It is not clear why you decided to combine the PHQ with the FSFI?

Depression is a known risk factor for sexual dysfunction in all age groups, gender, and populations. There is no reference depression in the introduction and it kind of surprises the reader that it was used. Include your reasoning on using the PHQ in the introduction.

Line 121: The information about validity and reliability assessed in 100 pts prior to conducting the study does not constitute a conventional psychometric evaluation process. How was validity (in the population under investigation) of the instruments determined? This could be viewed as a limitation.

Results

Line 150: “women in a stable monogamous and heterosexual relationship.” On what grounds are you able to make this statement? These determinants were not part of your sociodemographic information collected. How do you know this? Revision of this strong statement recommended.

Line 156: Why not report the 85.2% that are premenopausal? All the other sociodemographic characteristics, you report the higher %? Consistence is important.

Line 160: Are all partners tested in Nigeria? The options are only positive or negative partners. Are there no partners with their status unknown? Maybe alluded to this in the Introduction?

Line 164: Consistency recommended: (1743,59.6%) of the respondents. Rather 1743 (59.6%) of the respondents.

Line 169-172: Mention table 3 (consistency)

Line 177: “At least half of the respondents” = with scores of 76,66 and 50%? Review the sentence since it is way more than half?

Line 185: SD – it is the first time you use this abbreviation. What does it stand for? Standard deviation or sexual dysfunction? If it is sexual dysfunction, insert (SD) after the words “sexual dysfunction” in line 182. Or where you use it the first time in the article.

Page 20: Table 5 and 6 should be split in 2, each with its separate heading.

Line 194 -198: Consider mentioning the multivariates in the order as mentioned on table 5 &6:

Age, marital status, PHQ score >10, postmenopausal state, co-morbid medical conditions, HIV disclosure status, use of PI’s, duration of ART < 96 months, presence of opportunistic infection.

Highlight the factors in the same order for the sake of consistency: PHQ score >10, postmenopausal state, co-morbid medical conditions, HIV disclosure status, use of PI’s.

Discussion:

Line 220-221: “our setting and Sub-Saharan Africa” this is confusing. The study was only conducted in a single city in Nigeria as mentioned in line 207 – 210. Consider the removal of “and Sub-Saharan Africa”.

Line 225 -226: it will have more impact if you refer to the prevalence of Sexual dysfunction in women in general in Nigeria and highlight the high burden of sexual dysfunction in the HIV + population.

Line 228: is this referring to global frequency of abnormality or to the Nigerian setting?

Line 235: “More than half of the respondents” – review sentence as advised before.

Line 254: Menopausal status instead of “A menopausal state” maybe better to use.

Line 256: it is Possible – why not quote a study that prove the link with a hormonal imbalance in menopause with sexual dysfunction? To say it is possible, leave it open for speculation and I think science has established this association long ago.

Line 267 & 271: Ref 33 – Erectile dysfunction is associated with Hypertension and this study is done on women. Although it is a well-known sexual problem associated with hypertension, how relevant is this reference in an article reporting on findings in a female population? Do we have and specific sexual dysfunction associated in females with hypertension in the literature? Maybe refer to this systematic review for further research?

Santana, L.M., Perin, L., Lunelli, R. et al. Sexual Dysfunction in Women with Hypertension: a Systematic Review and Meta-analysis. Curr Hypertens Rep 21, 25 (2019). https://doi.org/10.1007/s11906-019-0925-z

Line 280: the word potency – what does it mean? Sexual interest and potency? Why not use one of the 6 domain words from the FSFI like desire?

Line 284-289: Discussion disclosure of HIV status and the fears surrounding it and referring the violence – please consider reference this statement?

The discussion address 5 of the 6 independently associated risk factors identified. There is no discussion about the PHQ score >10. As in the introduction – it is really not making sense why this has been included in the study methodology and mentioned in the results, but not put into perspective in the introduction and not discussed in the discussion? (It should be included, the evaluation of Depression is relevant to FSD). Address the omission in the 2 sections: Introduction and Discussion.

Consider consistency. Discuss the 6 factors in the exact order as before:

• PHQ score >10,

• postmenopausal state,

• co-morbid medical conditions,

• HIV disclosure status,

• use of PI’s.

Keeping the same order makes the reading and understanding of the findings so much easier to read and remember for a reader, especially if you have a non-academic reading the paper.

Conclusion

Line 304 – 305: why include Sub-Saharan Africa in this sentence?

6. PLOS authors have the option to publish the peer review history of their article (what does this mean?). If published, this will include your full peer review and any attached files.

Reviewer #1: **Yes: **dr padaruth ramlachan

Reviewer #2: **Yes: **Du Toit, MM

---

## [Author Response · Author response to Decision Letter 0]

15 Jan 2024

Dear Reviewers,

We would like to express my gratitude for the thorough review of my manuscript titled "Sexual dysfunction among Nigerian women living with HIV infection." We appreciate the valuable feedback provided, which has been instrumental in enhancing the quality and clarity of the manuscript.

We have carefully considered each comment and made necessary revisions to address the concerns raised. Find below the detailed responses.

Comments

Who is the corresponding author? An * is missing at one of the names.

Response:

We have included the * in the corresponding author’s name, see Line 2.

The corresponding author is Folahanmi Akinsolu

*Corresponding author

E-mail: folahanmi.tomiwa@gmail.com: Akinsolu.folahanmi@lcu.edu.ng

Introduction

Line 52: Review the need of including reference 1-3? Throughout the text, the first 3 references are never used. Consider removal and renumbering.

Response: 

I have reviewed the introduction, and I have added the three references in lines 49-52.

The introduction and increased access to antiretroviral therapy have changed human immunodeficiency virus (HIV) infection, a disease once considered a death sentence, to a chronic, manageable health condition. The widespread availability and use of potent combination antiretroviral therapy have dramatically reduced the morbidity and mortality of HIV infection globally and in sub-Saharan Africa. [1-4]

Line 58: “Numerous studies in sub-Saharan Africa” Reference 7 refers to a study done ins Tehran. Kindly motivate the use when referring to Sub-Saharan Africa.

Response:

The sentence has been revised. See line 58-60.

Numerous studies in sub-Saharan Africa and other parts of the world have examined the health-related quality of life in HIV-infected persons, with most studies emphasizing the physical, mental, and social aspects of quality of life while leaving out the sexual function component.

Line 60: Reference 7-21, should if not be 8-19?

Response:

Since the reference has been updated, the references have been revised. See line 58-60.

Numerous studies in sub-Saharan Africa and other parts of the world have examined the health-related quality of life in HIV-infected persons, with most studies emphasizing the physical, mental, and social aspects of quality of life while leaving out the sexual function component. [6-21]

Line 72: “But few studies”. Consider the grammar: but a few or only a few.

Response:

I have revised the statement extensively, see line 74-80.

There is still a lack of research and management of related issues in African countries [30] despite the many studies on sexual dysfunction in persons living with HIV that have been conducted in developed countries. [31, 32] Also, several studies have evaluated sexual dysfunction in HIV-positive men, [33] but few studies have examined the sexual function of women living with HIV/AIDS. [25]

Line 76: Relevance of ref 31-36: do you need all this reference? List is quite long.

Response:

I have revised the statement extensively, see line 74-80.

There is still a lack of research and management of related issues in African countries [30] despite the many studies on sexual dysfunction in persons living with HIV that have been conducted in developed countries. [31, 32] Also, several studies have evaluated sexual dysfunction in HIV-positive men,[33] but few studies have examined the sexual function of women living with HIV/AIDS.[25] Despite the high burden of HIV in Nigeria, an extensive literature search has revealed only three studies reporting on sexual dysfunction among HIV-positive women,[22-24] despite several studies on sexual dysfunction among women with chronic diseases, pregnancy, infertility, diabetes mellitus, or hypertension.[34-39]

Reference 31&33: these studies are about Hypertension and ED. Please motivate the use in a study on FSD.

Response:

I have rephrased the statement with the references, see line 74-80

There is still a lack of research and management of related issues in African countries [30] despite the many studies on sexual dysfunction in persons living with HIV that have been conducted in developed countries. [31, 32] Also, several studies have evaluated sexual dysfunction in HIV-positive men,[33] but few studies have examined the sexual function of women living with HIV/AIDS.[25] Despite the high burden of HIV in Nigeria, an extensive literature search has revealed only three studies reporting on sexual dysfunction among HIV-positive women,[22-24] despite several studies on sexual dysfunction among women with chronic diseases, pregnancy, infertility, diabetes mellitus, or hypertension.[34-39]

After reading the article, I suggest that you review the introduction. It needs to be more background information.

If sexologists read this work, they would be interested in the prevalence of sexual dysfunction in the general population in Nigeria. Mention the prevalence amongst the general population, followed by the prevalence in men, then in women. The aim of the study is to establish the prevalence in the female HIV+ population.

Address HIV burden in Nigeria – set the scene to the reader will give great context.

Include a short sentence of HIV treatment regimens offered in Nigeria.

Provide Examples of opportunistic infection and co-morbid medical conditions.

Response:

We have reviewed the introduction, see line 82-92

This pivotal study has a threefold objective. First, we aim to determine the prevalence of sexual dysfunction among HIV-positive women in Lagos, offering crucial quantitative insights into how widespread this issue is within this demographic. Next, we delve into the pattern of sexual dysfunction. Here, "pattern" refers to the detailed breakdown of sexual dysfunction as explored through the six domains of the Female Sexual Function Index (FSFI), which include desire, arousal, lubrication, orgasm, satisfaction, and pain. This comprehensive analysis allows us to understand not just the occurrence but the specific nature of sexual dysfunction experienced by these women. Lastly, we investigate the risk factors contributing to sexual dysfunction in this group, seeking to identify both the medical and psychosocial elements that might predispose these individuals to such challenges. By exploring these three critical areas, the study aims to provide a nuanced understanding of sexual dysfunction in HIV-positive women in Lagos, thereby laying the groundwork for more targeted and effective interventions.

Line 76: Conclude the Introduction section with the aim of the study clearly explained.

“The study was conducted to determine the

• Prevalence

• Pattern

• Risk factors of sexual dysfunction in HIV + Women in Lagos.

“Pattern” – elaborate on what you mean: The 6 domains tested with FSFI?

Response:

We have revised the concluding part of the introduction, see line 74-92

There is still a lack of research and management of related issues in African countries [30] despite the many studies on sexual dysfunction in persons living with HIV that have been conducted in developed countries. [31, 32] Also, several studies have evaluated sexual dysfunction in HIV-positive men,[33] but few studies have examined the sexual function of women living with HIV/AIDS.[25] Despite the high burden of HIV in Nigeria, an extensive literature search has revealed only three studies reporting on sexual dysfunction among HIV-positive women,[22-24] despite several studies on sexual dysfunction among women with chronic diseases, pregnancy, infertility, diabetes mellitus, or hypertension.[34-39] This study was conducted to determine the prevalence, pattern, and risk factors of sexual dysfunction among HIV-positive women in Lagos, Southwest Nigeria. This pivotal study has a threefold objective. First, we aim to determine the prevalence of sexual dysfunction among HIV-positive women in Lagos, offering crucial quantitative insights into how widespread this issue is within this demographic. Next, we delve into the pattern of sexual dysfunction. Here, "pattern" refers to the detailed breakdown of sexual dysfunction as explored through the six domains of the Female Sexual Function Index (FSFI), which include desire, arousal, lubrication, orgasm, satisfaction, and pain. This comprehensive analysis allows us to understand not just the occurrence but the specific nature of sexual dysfunction experienced by these women. Lastly, we investigate the risk factors contributing to sexual dysfunction in this group, seeking to identify both the medical and psychosocial elements that might predispose these individuals to such challenges. By exploring these three critical areas, the study aims to provide a nuanced understanding of sexual dysfunction in HIV-positive women in Lagos, thereby laying the groundwork for more targeted and effective interventions.

Methods

Line 98: Consider the word included instead of recruited.

Response:

We have included the word, “included” in the text, see line 102-103

A total of 3205 respondents randomly selected from the clinic attendance register were included in the study.

Line 100: What language version was used in both these measures? If English, how did you ascertain language proficiency in order to self-report?

Response:

We have included the language of the study tool, see line 116-117

The study tools were administered in English Language, and low-literate respondents were assisted in completing the forms by trained research assistants.

Line 102: It is not clear why you decided to combine the PHQ with the FSFI?

Depression is a known risk factor for sexual dysfunction in all age groups, gender, and populations. There is no reference depression in the introduction and it kind of surprises the reader that it was used. Include your reasoning on using the PHQ in the introduction.

Response:

Thank for giving us the opportunity to explain why we decided to combine PHQ and FSFI.

The decision to combine the Patient Health Questionnaire (PHQ) with the Female Sexual Function Index (FSFI) in our study was driven by the well-established connection between depression and sexual dysfunction across all age groups, genders, and populations. Although this link was not explicitly mentioned in the introduction, we recognize the importance of providing a clear rationale for our methodology. Depression significantly impacts an individual's overall well-being, including their sexual health.

Line 121: The information about validity and reliability assessed in 100 pts prior to conducting the study does not constitute a conventional psychometric evaluation process. How was validity (in the population under investigation) of the instruments determined? This could be viewed as a limitation.

Response:

We have explained how we determined the study tool validity and reliability assessed, see Line 134-137

The FSFI has been validated on a clinically diagnosed sample of women with female sexual disorders. [46] In our setting, [45] however, because the study was in a specific population, the validity and reliability in the study population were confirmed with 100 HIV-positive women. A Cronbach’s alpha score of 0.85 was obtained, indicating good reliability.

As a result, we have revised the study tools, data collection, and statistical analysis section of the study, see line 113-150.

Study tools, data collection, and statistical analysis 

A self-administered structured data collection form consisting of three sections, (A) Demographic characteristics, (B) Patient Depression Question (PHQ), and (C) Female Sexual Function Index, were used to collect data for the study. The study tools were administered in English Language, and low-literate respondents were assisted in completing the forms by trained research assistants. The respondents’ hospital numbers were written on each form to enable linkage to respondents’ medical records. The demographic section of the form captured respondents’ socioeconomic characteristics, medical and reproductive history, current and previous antiretroviral treatment, and recent laboratory results (CD4 cell count and viral load). The second section was the PHQ-9,[41, 42], a 9-item self-report questionnaire in which participants rated their feelings during the previous two weeks. Each question is scored 0 to 3 (0 = not at all, 1 = several days, 2 = more than half the days, and 3 = nearly every day), resulting in a 0 to 27. The nine items reflect the DSM-IV criteria for major depressive disorders. For this study, a score of 10 and above was considered depression. The last section is the Female Sexual Function Index (FSFI)[43, 44], a 19-item self-report instrument that scores six domains of sexual function in women in the previous four weeks. These domains include desire (two items, questions 1–2), arousal (four items, questions 3–6), lubrication (four items, questions 7–10), orgasm (three items, questions 11–13), satisfaction (three items, questions 14–16), and pain (three items, questions 17– 19). The total FSFI score was determined by the sum of the six domains, with the minimum and maximum scores possibly varying from 2.0 to 36.0. A score greater than 26.5 denoted a low level of sexual dysfunction, and a score of 26.5 or less indicated sexual dysfunction. Likewise, scores on desire, arousal, lubrication, orgasm, satisfaction, and pain, which are ≤ 3.6 (score range, 1.2–6), ≤ 3.9 (score range, 0–6), ≤ 3.6 (score range, 0–6), ≤ 3.6 (score range, 0–6), ≤ 3.6 (score range, 0–6) and ≤ 4.4 (score range, 0–6), respectively; all indicate sexual dysfunction related to subdimensions.[36, 43-45] The FSFI has been validated on a clinically diagnosed sample of women with female sexual disorders.[46] In our setting,[45] however, because the study was in a specific population, the validity and reliability in the study population were confirmed with 100 HIV-positive women. A Cronbach’s alpha score of 0.85 was obtained, indicating good reliability. 

The study results were presented as means and percentages. The total FSFI scores were categorized into dichotomous variables (normal sexual function and sexual dysfunction) using a cut-off point of 26.5.[44, 45] Frequency distributions were generated, and univariate analysis using relevant statistics was performed to identify factors associated with sexual dysfunction. Multivariate logistic regression was used to identify independent risk factors for sexual dysfunction while controlling for potential confounders, including the severity of HIV disease measured by CD4 count and HIV viral load, time since HIV diagnosis, duration of antiretroviral drug use, PHQ-9 score of ≥ 10, co-morbid state, education, employment status, marital status, age, and menopausal state. Variables were entered into the model if their P value on univariate analysis was ≤ 0.25 or less. The variable with the strongest association in the univariate model was estimated first, followed by others in descending order. In the analysis, the comparison group consisted of those with an FSFI score of > 26.5 (no sexual dysfunction). P < 0.05 was considered statistically significant. Odds Ratios (OR) and 95% Confidence Intervals (CI) for the OR were also calculated. All analyses were performed with SPSS statistical software package version 23.0 for Windows, SPSS Inc., Chicago, IL, USA.

Results

Line 150: “women in a stable monogamous and heterosexual relationship.” On what grounds are you able to make this statement? These determinants were not part of your sociodemographic information collected. How do you know this? Revision of this strong statement recommended.

Response:

Thank you for the observation, we have revised the statement, see line 161-164.

Out of 3205 HIV-positive women approached for participation in the study, a significant majority, 3009 (93.9%), consented and were either provided with the study forms or interviewed by research assistants. From this subset, 2926 (91.3%) women completed and returned the forms, thus providing a substantial and valuable dataset for our analysis.

Line 156: Why not report the 85.2% that are premenopausal? All the other sociodemographic characteristics, you report the higher %? Consistence is important.

Response:

Thank you, and we have revised the information to ensure consistence, see 170

2494 (85.2%) HIV-positive women were premenopausal.

Line 160: Are all partners tested in Nigeria? The options are only positive or negative partners. Are there no partners with their status unknown? Maybe alluded to this in the Introduction?

Response:

We appreciate the insightful query. While our study primarily focused on categorizing partners as either positive or negative, we acknowledge the importance of considering partners with unknown status in the context of HIV testing and disclosure. In the introduction, we provided background information on the challenges associated with disclosing HIV status, and we recognize that partners with unknown status play a significant role in this dynamic. Although it was not explicitly highlighted, we understand the relevance of including partners with unknown status in the discussion, as their presence can influence the dynamics of HIV disclosure and potential repercussions.

Line 164: Consistency recommended: (1743,59.6%) of the respondents. Rather 1743 (59.6%) of the respondents.

Response: 

Thank you, and we have revised the information to ensure consistence, see line 178.

Whereas 1743 (59.6%) of the respondents treated with ART have been receiving it for less than 96 months, the remaining 1182 (52.6%) had been receiving ART for 95 months or more. At the time of the interview, 509 (20.2%) women had opportunistic infections, and 564 (21.9%) had one or more co-morbid medical conditions.

Line 169-172: Mention table 3 (consistency)

Response:

Thank you, and we have revised the information to ensure consistence, see line 184-186.

The total score ranged from 0 – 25, with a mean of 5.6 ± 2.3. Most respondents had PHQ-9 scores of less than 10 (85.3%), and the remaining respondents (14.7%) had PHQ-9 scores of 10 and above. Only twenty-five respondents (0.8%) had a total score of 20 and above. (See Table 3)

Line 177: “At least half of the respondents” = with scores of 76,66 and 50%? Review the sentence since it is way more than half?

Response:

Thank you, and we have revised the information, see 192-194.

Over half of the respondents in this study reported experiencing dysfunction in several key sexual function domains: specifically, 76.8% experienced issues with desire, 66.0% with arousal, and 50.0% faced challenges with achieving orgasm.

Line 185: SD – it is the first time you use this abbreviation. What does it stand for? Standard deviation or sexual dysfunction? If it is sexual dysfunction, insert (SD) after the words “sexual dysfunction” in line 182. Or where you use it the first time in the article.

Response:

Thank you, and we have revised the information, see 199-204.

Table 5 shows the relationship between the women’s sociodemographic characteristics, co-morbid conditions, HIV characteristics, and sexual dysfunction. On univariate analysis, a significant association was found between women age ≥ 35 years (cOR: 1.4; 1.0 - 1.8), marital status (cOR:0.5;0.4 – 0.7), PHQ Score of ≥10 (cOR: 2.5; 1.6 – 4.0), postmenopausal status (cOR: 2.5; 1.1 – 5.6), co-morbid medical disorders (cOR: 1.8; 1.1 - 3.0) and sexual dysfunction. No statistically significant association was found between education (p=0.51), religion (p=0.47), employment status (p=0.31), and sexual dysfunction.

Page 20: Table 5 and 6 should be split in 2, each with its separate heading.

Response:

Thank you, and we have separated the tables, and inserted different headings, see pages 20-21.

Line 194 -198: Consider mentioning the multivariates in the order as mentioned on table 5 &6:

Age, marital status, PHQ score >10, postmenopausal state, co-morbid medical conditions, HIV disclosure status, use of PI’s, duration of ART < 96 months, presence of opportunistic infection.

Highlight the factors in the same order for the sake of consistency: PHQ score >10, postmenopausal state, co-morbid medical conditions, HIV disclosure status, use of PI’s. 

Response:

Thank you, and we have revised the information, see Lines 198-218

Table 5 shows the relationship between the women’s sociodemographic characteristics, co-morbid conditions, HIV characteristics, and sexual dysfunction. On univariate analysis, a significant association was found between women age ≥ 35 years (cOR: 1.4; 1.0 - 1.8), marital status (cOR:0.5;0.4 – 0.7), PHQ Score of ≥10 (cOR: 2.5; 1.6 – 4.0), postmenopausal status (cOR: 2.5; 1.1 – 5.6), co-morbid medical disorders (cOR: 1.8; 1.1 - 3.0) and sexual dysfunction. No statistically significant association was found between education (p=0.51), religion (p=0.47), employment status (p=0.31), and sexual dysfunction. 

The relationship between HIV-related factors and sexual dysfunction is summarized in Table 6. A statistically significant association was also found between HIV status disclosure (cOR:0.7; 0.6 – 0.97), use of protease inhibitor-based antiretroviral drug regimen (cOR: 1.6; 1.1 - 2.4), duration of HAART <96 months (cOR:1.5; 1.1 – 2.1), the presence of opportunistic infection (cOR: 1.6(1.1 – 2.4) and sexual dysfunction. Time since HIV diagnosis (p=0.18), CD4 count (p=0.8), Viral load (p=0.1), and partners’ positive HIV status (p=0.6) have no significant association with sexual dysfunction. 

After adjustment for potential confounding variables of age, PHQ score, time since HIV diagnosis, HIV status disclosure, antiretroviral drug regimen, presence of opportunistic infection, , and co-morbid medical conditions in the multivariate logistic regression model, sexual dysfunction in HIV-positive women had an independent statistically significant association with the postmenopausal state (aOR: 2.0; 1.4 – 4.1), PHQ score of ≥10 (aOR: 2.3; 1.7 – 3.2), co-morbid medical disorders (aOR: 1.8; 1.4 – 2.7), use of protease inhibitor-based regimen (aOR: 1.3; 1.2 - 2.1), and HIV status disclosure (aOR: 0.7; 0.6-0.8).

Discussion

Line 220-221: “our setting and Sub-Saharan Africa” this is confusing. The study was only conducted in a single city in Nigeria as mentioned in line 207 – 210. Consider the removal of “and Sub-Saharan Africa”.

Response:

Thank you, and we have revised the information, see line 235.

The sexual dysfunction rate of 71.4% in this study is within 61.0 – 89.2% reported in Nigerian studies.

Line 225 -226: it will have more impact if you refer to the prevalence of Sexual dysfunction in women in general in Nigeria and highlight the high burden of sexual dysfunction in the HIV + population.

Response:

Thank you, and we have revised the information, see Line 235-250.

The sexual dysfunction rate of 71.4% in this study is within 61.0 – 89.2% reported in Nigerian studies. 23-25 It is, however, higher than the rates reported among HIV-positive women in India[47] and Europe.[48, 49] Although female sexual dysfunction is a widespread health condition, controversy exists regarding the burden of female sexual dysfunction; vast differences are reported within and between countries. The differences may reflect medical and psychological factors, especially in socioeconomic; cultural; and ethnic differences; the definition used for each type of dysfunction; the type of study tool, the method of data collection; samples; and characteristics of populations studied.[22, 23, 25, 47-51] In addition, the observed higher prevalence in our setting may be due to cultural differences in the settings or reluctance of women in the sub-Saharan region to talk openly about sex and seek solutions for their sexual concerns.[11, 22, 23, 25, 47-51] The various explanations for the observed differences have also made generalization and comparison of studies challenging. This study highlights the significant burden of sexual dysfunction within the HIV-positive population in Nigeria, a concern that is particularly acute when considered against the backdrop of the already high prevalence (63%) of sexual dysfunction among women in general in the country [52]. This study used a validated and standardized tool and a specific population group to ensure generalization to our population group and context. This study confirmed that HIV infection is associated with high sexual dysfunction and thus deserves close attention.[22, 23, 25, 47-51]

Line 228: is this referring to global frequency of abnormality or to the Nigerian setting?

Response:

Thank you, and we have revised the information, see line 307-311.

As anticipated, the occurrence of abnormalities in specific sexual function domains is more pronounced in the Nigerian HIV-positive population studied than in the general Nigerian population. However, this frequency aligns with what has been reported in other studies focusing on HIV-positive women, indicating a consistent pattern of sexual dysfunction among HIV-positive women across different geographical and cultural contexts.[25, 48-50, 69]

Line 235: “More than half of the respondents” – review sentence as advised before.

Response:

Thank you, and we have revised the information, see line 316-318.

In this study, a significant majority of respondents experienced dysfunction in at least three specific sexual function domains, with 76.8% reporting issues with desire, 66.0% with arousal, and 50.0% with orgasm.

Line 254: Menopausal status instead of “A menopausal state” maybe better to use.

Response:

Thank you, and we have revised the information, see line 273-275.

Menopausal status was also found as a risk factor for sexual dysfunction in other studies involving women with known HIV status and the general population

Line 256: it is Possible – why not quote a study that prove the link with a hormonal imbalance in menopause with sexual dysfunction? To say it is possible, leave it open for speculation and I think science has established this association long ago.

Response:

Thank you, and we have revised the information, see line 267-275.

The association between changes in blood hormonal levels in women during menopause and a decline in sexual functioning is well-established in the scientific literature. Studies have consistently demonstrated that the hormonal imbalances experienced during menopause, particularly the reduction in estrogen and testosterone levels, are significantly linked to various aspects of sexual dysfunction. In a comprehensive review by Nappi and Lachowsky (2009) highlights the direct correlation between decreased estrogen levels and reduced vaginal lubrication, decreased libido, and other sexual health issues in menopausal women [58]. Menopausal status was also found as a risk factor for sexual dysfunction in other studies involving women with known HIV status and the general population.[11, 59-61]

Line 267 & 271: Ref 33 – Erectile dysfunction is associated with Hypertension and this study is done on women. Although it is a well-known sexual problem associated with hypertension, how relevant is this reference in an article reporting on findings in a female population? Do we have and specific sexual dysfunction associated in females with hypertension in the literature? Maybe refer to this systematic review for further research?

Santana, L.M., Perin, L., Lunelli, R. et al. Sexual Dysfunction in Women with Hypertension: a Systematic Review and Meta-analysis. Curr Hypertens Rep 21, 25 (2019). https://doi.org/10.1007/s11906-019-0925-z

Response:

Thank you, and we have revised the information, see line 285-296.

Hypertension and diabetes are prevalent comorbidities in our environment, particularly among the HIV patient population, [64] and 61-88% of patients with these diseases experiencing sexual dysfunction.[11, 25, 37, 65] While erectile dysfunction is a well-recognized consequences of hypertension in men, women with hypertension also face significant sexual health challenges, including reduced sexual desire, arousal issues, and overall dissatisfaction with sexual life. The conclusive evidence of this association between hypertension and various forms of sexual dysfunction in women is well-documented in a systematic review and meta-analysis by Santana, L.M., Perin, L., Lunelli, R. et al. (2019).[66] Given the high prevalence of sexual dysfunction among patients with these comorbid conditions, it is not surprising that co-morbid medical conditions retained an independent statistical association with sexual dysfunction, even after controlling for other possible confounders. This association is consistent with findings from other studies in our environment.

Line 280: the word potency – what does it mean? Sexual interest and potency? Why not use one of the 6 domain words from the FSFI like desire?

Response:

Thank you, and we have revised the information, see line 333-339.

Castelo-Branco et al.,[71] in their study among healthy women living in Santiago de Chile, found desire disorder (38.0%) the most common, followed by arousal disorder (32.0%), pain disorder (33%), and orgasm disorder (25.0%). In a study by Kadri and colleagues[73] among women living in Morocco, desire disorder was reported by 18%, orgasm disorder by 12%, arousal disorder by 8%, and pain disorder by 8%. The inference that can be made from the above is that while the severity of specific domain dysfunction may be worse in HIV-positive women, their distribution and pattern of occurrence are like those in HIV-positive and HIV-negative women.[11, 22, 23, 25, 61, 73]

Line 284-289: Discussion disclosure of HIV status and the fears surrounding it and referring the violence – please consider reference this statement?

Response:

Thank you, and we have revised the information, see line 326-332.

Castelo-Branco et al.,[71] in their study among healthy women living in Santiago de Chile, found desire disorder (38.0%) the most common, followed by arousal disorder (32.0%), pain disorder (33%), and orgasm disorder (25.0%). In a study by Kadri and colleagues[73] among women living in Morocco, desire disorder was reported by 18%, orgasm disorder by 12%, arousal disorder by 8%, and pain disorder by 8%. The inference that can be made from the above is that while the severity of specific domain dysfunction may be worse in HIV-positive women, their distribution and pattern of occurrence are like those in HIV-positive and HIV-negative women.[11, 22, 23, 25, 61, 73]

The discussion address 5 of the 6 independently associated risk factors identified. There is no discussion about the PHQ score >10. As in the introduction – it is really not making sense why this has been included in the study methodology and mentioned in the results, but not put into perspective in the introduction and not discussed in the discussion? (It should be included, the evaluation of Depression is relevant to FSD). Address the omission in the 2 sections: Introduction and Discussion.

Consider consistency. Discuss the 6 factors in the exact order as before:

• PHQ score >10,

• postmenopausal state,

• co-morbid medical conditions,

• HIV disclosure status,

• use of PI’s.

Keeping the same order makes the reading and understanding of the findings so much easier to read and remember for a reader, especially if you have a non-academic reading the paper.

Response:

Thank you, and we have revised the information, see line 252-263.

The result PHQ-9 score of ≥10 is (cOR) of 2.5 (95% CI: 1.6 – 4.0) shows that sexual dysfunction is associated with depressive symptoms. A study by Atlantis & Sullivan (2012) found a strong link between depression and sexual dysfunction, suggesting that the presence of depressive symptoms can significantly increase the likelihood of experiencing difficulties with sexual function.[53] A study by Clayton et al. (2014) found similar results, noting that depression was a significant predictor of sexual dysfunction in women, and suggested that treatment of depressive symptoms could have beneficial effects on sexual function.[54] The study by Mathew & Weinman (2009) also found a strong association between depression and sexual dysfunction, further supporting the findings of this study.[55] However, while these results are compelling, it's essential to consider them in the context of potential confounders and the complexity of sexual health. Factors such as medication side effects, relationship issues, and other psychological or physical health conditions can also significantly impact sexual function. The relationship between depression and sexual dysfunction is likely to be bidirectional and influenced by a range of other factors.[56, 57]

Conclusion

Line 304 – 305: why include Sub-Saharan Africa in this sentence?

Response:

Thank you, and we have revised the information, see line 355-363.

Sexual dysfunction is a significant public health challenge among HIV-positive women in Nigeria, where HIV is endemic. The sexual dysfunction rate of 71.4% confirms that it is a significant challenge in our setting. Sexual dysfunction in the six domains ranged from 38.8% in the satisfaction domain to 76.8% in the desire domain. Six factors are significantly associated with sexual dysfunction: menopausal state, PHQ score of 10 and above, opportunistic infection, co-morbid medical condition, use of protease-based antiretroviral regimen, and non-disclosure of HIV status. There is a need to sensitize and train healthcare workers to assess for sexual challenges during routine programmatic consultations. The national HIV programs should, as a matter of urgency, include screening and treatment of sexual dysfunctions among HIV-positive women in the country’s treatment guidelines.

---

## [Editor Report · Decision Letter 1]

15 Mar 2024

PONE-D-23-24987R1Sexual Dysfunction among Nigerian Women Living with HIV InfectionPLOS ONE  Dear Dr. Akinsolu,

Thank you for submitting your manuscript to PLOS ONE. After careful consideration, we feel that it has merit but does not fully meet PLOS ONE’s publication criteria as it currently stands. Therefore, we invite you to submit a revised version of the manuscript that addresses the points raised during the review process. ==============================

Please consider the following comments:

Who is the corresponding author? An * is missing at one of the names.

Introduction:

Line 52: Review the need of including reference 1-3? Throughout the text, the first 3 references are never used. Consider removal and renumbering.

Line 58: “Numerous studies in sub-Saharan Africa” Reference 7 refers to a study done ins Tehran. Kindly motivate the use when referring to Sub-Saharan Africa.

Line 60: Reference 7-21, should if not be 8-19?

Line 72: “But few studies”. Consider the grammar: but a few or only a few.

Line 76: Relevance of ref 31-36: do you need all this references? List is quite long.

Reference 31& 33: these studies are about Hypertension and ED. Please motivate the use in a study on FSD.

After reading the article, I suggest that you review the introduction. It needs to be more background information.

If sexologists read this work, they would be interested in the prevalence of sexual dysfunction in the general population in Nigeria. Mention the prevalence amongst the general population, followed by the prevalence in men, then in women. The aim of the study is to establish the prevalence in the female HIV+ population.

Address HIV burden in Nigeria – set the scene to the reader will give great context.

Include a short sentence of HIV treatment regimens offered in Nigeria.

Provide Examples of opportunistic infection and co-morbid medical conditions.

Line 76: Conclude the Introduction section with the aim of the study clearly explained.

“The study was conducted to determine the

• Prevalence

• Pattern

• Risk factors of sexual dysfunction in HIV + Women in Lagos.

“Pattern” – elaborate on what you mean: The 6 domains tested with FSFI?

Methods

Line 98: Consider the word included instead of recruited.

Line 100: What language version was used in both these measures? If English, how did you ascertain language proficiency in order to self-report?

Line 102: It is not clear why you decided to combine the PHQ with the FSFI?

Depression is a known risk factor for sexual dysfunction in all age groups, gender, and populations. There is no reference depression in the introduction and it kind of surprises the reader that it was used. Include your reasoning on using the PHQ in the introduction.

Line 121: The information about validity and reliability assessed in 100 pts prior to conducting the study does not constitute a conventional psychometric evaluation process. How was validity (in the population under investigation) of the instruments determined? This could be viewed as a limitation.

Results

Line 150: “women in a stable monogamous and heterosexual relationship.” On what grounds are you able to make this statement? These determinants were not part of your sociodemographic information collected. How do you know this? Revision of this strong statement recommended.

Line 156: Why not report the 85.2% that are premenopausal? All the other sociodemographic characteristics, you report the higher %? Consistence is important.

Line 160: Are all partners tested in Nigeria? The options are only positive or negative partners. Are there no partners with their status unknown? Maybe alluded to this in the Introduction?

Line 164: Consistency recommended: (1743,59.6%) of the respondents. Rather 1743 (59.6%) of the respondents.

Line 169-172: Mention table 3 (consistency)

Line 177: “At least half of the respondents” = with scores of 76,66 and 50%? Review the sentence since it is way more than half?

Line 185: SD – it is the first time you use this abbreviation. What does it stand for? Standard deviation or sexual dysfunction? If it is sexual dysfunction, insert (SD) after the words “sexual dysfunction” in line 182. Or where you use it the first time in the article.

Page 20: Table 5 and 6 should be split in 2, each with its separate heading.

Line 194 -198: Consider mentioning the multivariates in the order as mentioned on table 5 &6:

Age, marital status, PHQ score >10, postmenopausal state, co-morbid medical conditions, HIV disclosure status, use of PI’s, duration of ART < 96 months, presence of opportunistic infection.

Highlight the factors in the same order for the sake of consistency: PHQ score >10, postmenopausal state, co-morbid medical conditions, HIV disclosure status, use of PI’s.

Discussion:

Line 220-221: “our setting and Sub-Saharan Africa” this is confusing. The study was only conducted in a single city in Nigeria as mentioned in line 207 – 210. Consider the removal of “and Sub-Saharan Africa”.

Line 225 -226: it will have more impact if you refer to the prevalence of Sexual dysfunction in women in general in Nigeria and highlight the high burden of sexual dysfunction in the HIV + population.

Line 228: is this referring to global frequency of abnormality or to the Nigerian setting?

Line 235: “More than half of the respondents” – review sentence as advised before.

Line 254: Menopausal status instead of “A menopausal state” maybe better to use.

Line 256: it is Possible – why not quote a study that prove the link with a hormonal imbalance in menopause with sexual dysfunction? To say it is possible, leave it open for speculation and I think science has established this association long ago.

Line 267 & 271: Ref 33 – Erectile dysfunction is associated with Hypertension and this study is done on women. Although it is a well-known sexual problem associated with hypertension, how relevant is this reference in an article reporting on findings in a female population? Do we have and specific sexual dysfunction associated in females with hypertension in the literature? Maybe refer to this systematic review for further research?

Santana, L.M., Perin, L., Lunelli, R. et al. Sexual Dysfunction in Women with Hypertension: a Systematic Review and Meta-analysis. Curr Hypertens Rep 21, 25 (2019). https://doi.org/10.1007/s11906-019-0925-z

Line 280: the word potency – what does it mean? Sexual interest and potency? Why not use one of the 6 domain words from the FSFI like desire?

Line 284-289: Discussion disclosure of HIV status and the fears surrounding it and referring the violence – please consider reference this statement?

The discussion address 5 of the 6 independently associated risk factors identified. There is no discussion about the PHQ score >10. As in the introduction – it is really not making sense why this has been included in the study methodology and mentioned in the results, but not put into perspective in the introduction and not discussed in the discussion? (It should be included, the evaluation of Depression is relevant to FSD). Address the omission in the 2 sections: Introduction and Discussion.

Consider consistency. Discuss the 6 factors in the exact order as before:

• PHQ score >10,

• postmenopausal state,

• co-morbid medical conditions,

• HIV disclosure status,

• use of PI’s.

Keeping the same order makes the reading and understanding of the findings so much easier to read and remember for a reader, especially if you have a non-academic reading the paper.

Conclusion

Line 304 – 305: why include Sub-Saharan Africa in this sentence?ata. 

Please submit your revised manuscript by Apr 29 2024 11:59PM. If you will need more time than this to complete your revisions, please reply to this message or contact the journal office at plosone@plos.org. Please include the following items when submitting your revised manuscript:A rebuttal letter that responds to each point raised by the academic editor and reviewer(s). You should upload this letter as a separate file labeled 'Response to Reviewers'.A marked-up copy of your manuscript that highlights changes made to the original version. You should upload this as a separate file labeled 'Revised Manuscript with Track Changes'.An unmarked version of your revised paper without tracked changes. You should upload this as a separate file labeled 'Manuscript'.If applicable, we recommend that you deposit your laboratory protocols in protocols.io to enhance the reproducibility of your results. Protocols.io assigns your protocol its own identifier (DOI) so that it can be cited independently in the future. For instructions see: https://journals.plos.org/plosone/s/submission-guidelines#loc-laboratory-protocols. Additionally, PLOS ONE offers an option for publishing peer-reviewed Lab Protocol articles, which describe protocols hosted on protocols.io. Read more information on sharing protocols at https://plos.org/protocols?utm_medium=editorial-email&utm_source=authorletters&utm_campaign=protocols.

We look forward to receiving your revised manuscript.

Kind regards,

Deidre Pretorius, PhD

Academic Editor

PLOS ONE

Journal Requirements:

Additional Editor Comments:

Please see the reviewer feedback

---

## [Author Response · Author response to Decision Letter 1]

17 Mar 2024

Dear Reviewers,

We would like to express my gratitude for the thorough review of my manuscript titled "Sexual dysfunction among Nigerian women living with HIV infection." We appreciate the valuable feedback provided, which has been instrumental in enhancing the quality and clarity of the manuscript.

We have carefully considered each comment and made necessary revisions to address the concerns raised. Find below the detailed responses.

Comments

Who is the corresponding author? An * is missing at one of the names.

Response:

We have included the * in the corresponding author’s name, see Line 2.

The corresponding author is Folahanmi Akinsolu

*Corresponding author

E-mail: folahanmi.tomiwa@gmail.com: Akinsolu.folahanmi@lcu.edu.ng

Introduction

Line 52: Review the need of including reference 1-3? Throughout the text, the first 3 references are never used. Consider removal and renumbering.

Response: 

I have reviewed the introduction, and I have added the three references in lines 49-52.

The introduction and increased access to antiretroviral therapy have changed human immunodeficiency virus (HIV) infection, a disease once considered a death sentence, to a chronic, manageable health condition. The widespread availability and use of potent combination antiretroviral therapy have dramatically reduced the morbidity and mortality of HIV infection globally and in sub-Saharan Africa. [1-4]

Line 58: “Numerous studies in sub-Saharan Africa” Reference 7 refers to a study done ins Tehran. Kindly motivate the use when referring to Sub-Saharan Africa.

Response:

The sentence has been revised. See line 58-60.

Numerous studies in sub-Saharan Africa and other parts of the world have examined the health-related quality of life in HIV-infected persons, with most studies emphasizing the physical, mental, and social aspects of quality of life while leaving out the sexual function component.

Line 60: Reference 7-21, should if not be 8-19?

Response:

Since the reference has been updated, the references have been revised. See line 58-60.

Numerous studies in sub-Saharan Africa and other parts of the world have examined the health-related quality of life in HIV-infected persons, with most studies emphasizing the physical, mental, and social aspects of quality of life while leaving out the sexual function component. [6-21]

Line 72: “But few studies”. Consider the grammar: but a few or only a few.

Response:

I have revised the statement extensively, see line 74-80.

There is still a lack of research and management of related issues in African countries [30] despite the many studies on sexual dysfunction in persons living with HIV that have been conducted in developed countries. [31, 32] Also, several studies have evaluated sexual dysfunction in HIV-positive men, [33] but few studies have examined the sexual function of women living with HIV/AIDS. [25]

Line 76: Relevance of ref 31-36: do you need all this reference? List is quite long.

Response:

I have revised the statement extensively, see line 74-80.

There is still a lack of research and management of related issues in African countries [30] despite the many studies on sexual dysfunction in persons living with HIV that have been conducted in developed countries. [31, 32] Also, several studies have evaluated sexual dysfunction in HIV-positive men,[33] but few studies have examined the sexual function of women living with HIV/AIDS.[25] Despite the high burden of HIV in Nigeria, an extensive literature search has revealed only three studies reporting on sexual dysfunction among HIV-positive women,[22-24] despite several studies on sexual dysfunction among women with chronic diseases, pregnancy, infertility, diabetes mellitus, or hypertension.[34-39]

Reference 31&33: these studies are about Hypertension and ED. Please motivate the use in a study on FSD.

Response:

I have rephrased the statement with the references, see line 74-80

There is still a lack of research and management of related issues in African countries [30] despite the many studies on sexual dysfunction in persons living with HIV that have been conducted in developed countries. [31, 32] Also, several studies have evaluated sexual dysfunction in HIV-positive men,[33] but few studies have examined the sexual function of women living with HIV/AIDS.[25] Despite the high burden of HIV in Nigeria, an extensive literature search has revealed only three studies reporting on sexual dysfunction among HIV-positive women,[22-24] despite several studies on sexual dysfunction among women with chronic diseases, pregnancy, infertility, diabetes mellitus, or hypertension.[34-39]

After reading the article, I suggest that you review the introduction. It needs to be more background information.

If sexologists read this work, they would be interested in the prevalence of sexual dysfunction in the general population in Nigeria. Mention the prevalence amongst the general population, followed by the prevalence in men, then in women. The aim of the study is to establish the prevalence in the female HIV+ population.

Address HIV burden in Nigeria – set the scene to the reader will give great context.

Include a short sentence of HIV treatment regimens offered in Nigeria.

Provide Examples of opportunistic infection and co-morbid medical conditions.

Response:

We have reviewed the introduction, see line 82-92

This pivotal study has a threefold objective. First, we aim to determine the prevalence of sexual dysfunction among HIV-positive women in Lagos, offering crucial quantitative insights into how widespread this issue is within this demographic. Next, we delve into the pattern of sexual dysfunction. Here, "pattern" refers to the detailed breakdown of sexual dysfunction as explored through the six domains of the Female Sexual Function Index (FSFI), which include desire, arousal, lubrication, orgasm, satisfaction, and pain. This comprehensive analysis allows us to understand not just the occurrence but the specific nature of sexual dysfunction experienced by these women. Lastly, we investigate the risk factors contributing to sexual dysfunction in this group, seeking to identify both the medical and psychosocial elements that might predispose these individuals to such challenges. By exploring these three critical areas, the study aims to provide a nuanced understanding of sexual dysfunction in HIV-positive women in Lagos, thereby laying the groundwork for more targeted and effective interventions.

Line 76: Conclude the Introduction section with the aim of the study clearly explained.

“The study was conducted to determine the

• Prevalence

• Pattern

• Risk factors of sexual dysfunction in HIV + Women in Lagos.

“Pattern” – elaborate on what you mean: The 6 domains tested with FSFI?

Response:

We have revised the concluding part of the introduction, see line 74-92

There is still a lack of research and management of related issues in African countries [30] despite the many studies on sexual dysfunction in persons living with HIV that have been conducted in developed countries. [31, 32] Also, several studies have evaluated sexual dysfunction in HIV-positive men,[33] but few studies have examined the sexual function of women living with HIV/AIDS.[25] Despite the high burden of HIV in Nigeria, an extensive literature search has revealed only three studies reporting on sexual dysfunction among HIV-positive women,[22-24] despite several studies on sexual dysfunction among women with chronic diseases, pregnancy, infertility, diabetes mellitus, or hypertension.[34-39] This study was conducted to determine the prevalence, pattern, and risk factors of sexual dysfunction among HIV-positive women in Lagos, Southwest Nigeria. This pivotal study has a threefold objective. First, we aim to determine the prevalence of sexual dysfunction among HIV-positive women in Lagos, offering crucial quantitative insights into how widespread this issue is within this demographic. Next, we delve into the pattern of sexual dysfunction. Here, "pattern" refers to the detailed breakdown of sexual dysfunction as explored through the six domains of the Female Sexual Function Index (FSFI), which include desire, arousal, lubrication, orgasm, satisfaction, and pain. This comprehensive analysis allows us to understand not just the occurrence but the specific nature of sexual dysfunction experienced by these women. Lastly, we investigate the risk factors contributing to sexual dysfunction in this group, seeking to identify both the medical and psychosocial elements that might predispose these individuals to such challenges. By exploring these three critical areas, the study aims to provide a nuanced understanding of sexual dysfunction in HIV-positive women in Lagos, thereby laying the groundwork for more targeted and effective interventions.

Methods

Line 98: Consider the word included instead of recruited.

Response:

We have included the word, “included” in the text, see line 102-103

A total of 3205 respondents randomly selected from the clinic attendance register were included in the study.

Line 100: What language version was used in both these measures? If English, how did you ascertain language proficiency in order to self-report?

Response:

We have included the language of the study tool, see line 116-117

The study tools were administered in English Language, and low-literate respondents were assisted in completing the forms by trained research assistants.

Line 102: It is not clear why you decided to combine the PHQ with the FSFI?

Depression is a known risk factor for sexual dysfunction in all age groups, gender, and populations. There is no reference depression in the introduction and it kind of surprises the reader that it was used. Include your reasoning on using the PHQ in the introduction.

Response:

Thank for giving us the opportunity to explain why we decided to combine PHQ and FSFI.

The decision to combine the Patient Health Questionnaire (PHQ) with the Female Sexual Function Index (FSFI) in our study was driven by the well-established connection between depression and sexual dysfunction across all age groups, genders, and populations. Although this link was not explicitly mentioned in the introduction, we recognize the importance of providing a clear rationale for our methodology. Depression significantly impacts an individual's overall well-being, including their sexual health.

Line 121: The information about validity and reliability assessed in 100 pts prior to conducting the study does not constitute a conventional psychometric evaluation process. How was validity (in the population under investigation) of the instruments determined? This could be viewed as a limitation.

Response:

We have explained how we determined the study tool validity and reliability assessed, see Line 134-137

The FSFI has been validated on a clinically diagnosed sample of women with female sexual disorders. [46] In our setting, [45] however, because the study was in a specific population, the validity and reliability in the study population were confirmed with 100 HIV-positive women. A Cronbach’s alpha score of 0.85 was obtained, indicating good reliability.

As a result, we have revised the study tools, data collection, and statistical analysis section of the study, see line 113-150.

Study tools, data collection, and statistical analysis 

A self-administered structured data collection form consisting of three sections, (A) Demographic characteristics, (B) Patient Depression Question (PHQ), and (C) Female Sexual Function Index, were used to collect data for the study. The study tools were administered in English Language, and low-literate respondents were assisted in completing the forms by trained research assistants. The respondents’ hospital numbers were written on each form to enable linkage to respondents’ medical records. The demographic section of the form captured respondents’ socioeconomic characteristics, medical and reproductive history, current and previous antiretroviral treatment, and recent laboratory results (CD4 cell count and viral load). The second section was the PHQ-9,[41, 42], a 9-item self-report questionnaire in which participants rated their feelings during the previous two weeks. Each question is scored 0 to 3 (0 = not at all, 1 = several days, 2 = more than half the days, and 3 = nearly every day), resulting in a 0 to 27. The nine items reflect the DSM-IV criteria for major depressive disorders. For this study, a score of 10 and above was considered depression. The last section is the Female Sexual Function Index (FSFI)[43, 44], a 19-item self-report instrument that scores six domains of sexual function in women in the previous four weeks. These domains include desire (two items, questions 1–2), arousal (four items, questions 3–6), lubrication (four items, questions 7–10), orgasm (three items, questions 11–13), satisfaction (three items, questions 14–16), and pain (three items, questions 17– 19). The total FSFI score was determined by the sum of the six domains, with the minimum and maximum scores possibly varying from 2.0 to 36.0. A score greater than 26.5 denoted a low level of sexual dysfunction, and a score of 26.5 or less indicated sexual dysfunction. Likewise, scores on desire, arousal, lubrication, orgasm, satisfaction, and pain, which are ≤ 3.6 (score range, 1.2–6), ≤ 3.9 (score range, 0–6), ≤ 3.6 (score range, 0–6), ≤ 3.6 (score range, 0–6), ≤ 3.6 (score range, 0–6) and ≤ 4.4 (score range, 0–6), respectively; all indicate sexual dysfunction related to subdimensions.[36, 43-45] The FSFI has been validated on a clinically diagnosed sample of women with female sexual disorders.[46] In our setting,[45] however, because the study was in a specific population, the validity and reliability in the study population were confirmed with 100 HIV-positive women. A Cronbach’s alpha score of 0.85 was obtained, indicating good reliability. 

The study results were presented as means and percentages. The total FSFI scores were categorized into dichotomous variables (normal sexual function and sexual dysfunction) using a cut-off point of 26.5.[44, 45] Frequency distributions were generated, and univariate analysis using relevant statistics was performed to identify factors associated with sexual dysfunction. Multivariate logistic regression was used to identify independent risk factors for sexual dysfunction while controlling for potential confounders, including the severity of HIV disease measured by CD4 count and HIV viral load, time since HIV diagnosis, duration of antiretroviral drug use, PHQ-9 score of ≥ 10, co-morbid state, education, employment status, marital status, age, and menopausal state. Variables were entered into the model if their P value on univariate analysis was ≤ 0.25 or less. The variable with the strongest association in the univariate model was estimated first, followed by others in descending order. In the analysis, the comparison group consisted of those with an FSFI score of > 26.5 (no sexual dysfunction). P < 0.05 was considered statistically significant. Odds Ratios (OR) and 95% Confidence Intervals (CI) for the OR were also calculated. All analyses were performed with SPSS statistical software package version 23.0 for Windows, SPSS Inc., Chicago, IL, USA.

Results

Line 150: “women in a stable monogamous and heterosexual relationship.” On what grounds are you able to make this statement? These determinants were not part of your sociodemographic information collected. How do you know this? Revision of this strong statement recommended.

Response:

Thank you for the observation, we have revised the statement, see line 161-164.

Out of 3205 HIV-positive women approached for participation in the study, a significant majority, 3009 (93.9%), consented and were either provided with the study forms or interviewed by research assistants. From this subset, 2926 (91.3%) women completed and returned the forms, thus providing a substantial and valuable dataset for our analysis.

Line 156: Why not report the 85.2% that are premenopausal? All the other sociodemographic characteristics, you report the higher %? Consistence is important.

Response:

Thank you, and we have revised the information to ensure consistence, see 170

2494 (85.2%) HIV-positive women were premenopausal.

Line 160: Are all partners tested in Nigeria? The options are only positive or negative partners. Are there no partners with their status unknown? Maybe alluded to this in the Introduction?

Response:

We appreciate the insightful query. While our study primarily focused on categorizing partners as either positive or negative, we acknowledge the importance of considering partners with unknown status in the context of HIV testing and disclosure. In the introduction, we provided background information on the challenges associated with disclosing HIV status, and we recognize that partners with unknown status play a significant role in this dynamic. Although it was not explicitly highlighted, we understand the relevance of including partners with unknown status in the discussion, as their presence can influence the dynamics of HIV disclosure and potential repercussions.

Line 164: Consistency recommended: (1743,59.6%) of the respondents. Rather 1743 (59.6%) of the respondents.

Response: 

Thank you, and we have revised the information to ensure consistence, see line 178.

Whereas 1743 (59.6%) of the respondents treated with ART have been receiving it for less than 96 months, the remaining 1182 (52.6%) had been receiving ART for 95 months or more. At the time of the interview, 509 (20.2%) women had opportunistic infections, and 564 (21.9%) had one or more co-morbid medical conditions.

Line 169-172: Mention table 3 (consistency)

Response:

Thank you, and we have revised the information to ensure consistence, see line 184-186.

The total score ranged from 0 – 25, with a mean of 5.6 ± 2.3. Most respondents had PHQ-9 scores of less than 10 (85.3%), and the remaining respondents (14.7%) had PHQ-9 scores of 10 and above. Only twenty-five respondents (0.8%) had a total score of 20 and above. (See Table 3)

Line 177: “At least half of the respondents” = with scores of 76,66 and 50%? Review the sentence since it is way more than half?

Response:

Thank you, and we have revised the information, see 192-194.

Over half of the respondents in this study reported experiencing dysfunction in several key sexual function domains: specifically, 76.8% experienced issues with desire, 66.0% with arousal, and 50.0% faced challenges with achieving orgasm.

Line 185: SD – it is the first time you use this abbreviation. What does it stand for? Standard deviation or sexual dysfunction? If it is sexual dysfunction, insert (SD) after the words “sexual dysfunction” in line 182. Or where you use it the first time in the article.

Response:

Thank you, and we have revised the information, see 199-204.

Table 5 shows the relationship between the women’s sociodemographic characteristics, co-morbid conditions, HIV characteristics, and sexual dysfunction. On univariate analysis, a significant association was found between women age ≥ 35 years (cOR: 1.4; 1.0 - 1.8), marital status (cOR:0.5;0.4 – 0.7), PHQ Score of ≥10 (cOR: 2.5; 1.6 – 4.0), postmenopausal status (cOR: 2.5; 1.1 – 5.6), co-morbid medical disorders (cOR: 1.8; 1.1 - 3.0) and sexual dysfunction. No statistically significant association was found between education (p=0.51), religion (p=0.47), employment status (p=0.31), and sexual dysfunction.

Page 20: Table 5 and 6 should be split in 2, each with its separate heading.

Response:

Thank you, and we have separated the tables, and inserted different headings, see pages 20-21.

Line 194 -198: Consider mentioning the multivariates in the order as mentioned on table 5 &6:

Age, marital status, PHQ score >10, postmenopausal state, co-morbid medical conditions, HIV disclosure status, use of PI’s, duration of ART < 96 months, presence of opportunistic infection.

Highlight the factors in the same order for the sake of consistency: PHQ score >10, postmenopausal state, co-morbid medical conditions, HIV disclosure status, use of PI’s. 

Response:

Thank you, and we have revised the information, see Lines 198-218

Table 5 shows the relationship between the women’s sociodemographic characteristics, co-morbid conditions, HIV characteristics, and sexual dysfunction. On univariate analysis, a significant association was found between women age ≥ 35 years (cOR: 1.4; 1.0 - 1.8), marital status (cOR:0.5;0.4 – 0.7), PHQ Score of ≥10 (cOR: 2.5; 1.6 – 4.0), postmenopausal status (cOR: 2.5; 1.1 – 5.6), co-morbid medical disorders (cOR: 1.8; 1.1 - 3.0) and sexual dysfunction. No statistically significant association was found between education (p=0.51), religion (p=0.47), employment status (p=0.31), and sexual dysfunction. 

The relationship between HIV-related factors and sexual dysfunction is summarized in Table 6. A statistically significant association was also found between HIV status disclosure (cOR:0.7; 0.6 – 0.97), use of protease inhibitor-based antiretroviral drug regimen (cOR: 1.6; 1.1 - 2.4), duration of HAART <96 months (cOR:1.5; 1.1 – 2.1), the presence of opportunistic infection (cOR: 1.6(1.1 – 2.4) and sexual dysfunction. Time since HIV diagnosis (p=0.18), CD4 count (p=0.8), Viral load (p=0.1), and partners’ positive HIV status (p=0.6) have no significant association with sexual dysfunction. 

After adjustment for potential confounding variables of age, PHQ score, time since HIV diagnosis, HIV status disclosure, antiretroviral drug regimen, presence of opportunistic infection, , and co-morbid medical conditions in the multivariate logistic regression model, sexual dysfunction in HIV-positive women had an independent statistically significant association with the postmenopausal state (aOR: 2.0; 1.4 – 4.1), PHQ score of ≥10 (aOR: 2.3; 1.7 – 3.2), co-morbid medical disorders (aOR: 1.8; 1.4 – 2.7), use of protease inhibitor-based regimen (aOR: 1.3; 1.2 - 2.1), and HIV status disclosure (aOR: 0.7; 0.6-0.8).

Discussion

Line 220-221: “our setting and Sub-Saharan Africa” this is confusing. The study was only conducted in a single city in Nigeria as mentioned in line 207 – 210. Consider the removal of “and Sub-Saharan Africa”.

Response:

Thank you, and we have revised the information, see line 235.

The sexual dysfunction rate of 71.4% in this study is within 61.0 – 89.2% reported in Nigerian studies.

Line 225 -226: it will have more impact if you refer to the prevalence of Sexual dysfunction in women in general in Nigeria and highlight the high burden of sexual dysfunction in the HIV + population.

Response:

Thank you, and we have revised the information, see Line 235-250.

The sexual dysfunction rate of 71.4% in this study is within 61.0 – 89.2% reported in Nigerian studies. 23-25 It is, however, higher than the rates reported among HIV-positive women in India[47] and Europe.[48, 49] Although female sexual dysfunction is a widespread health condition, controversy exists regarding the burden of female sexual dysfunction; vast differences are reported within and between countries. The differences may reflect medical and psychological factors, especially in socioeconomic; cultural; and ethnic differences; the definition used for each type of dysfunction; the type of study tool, the method of data collection; samples; and characteristics of populations studied.[22, 23, 25, 47-51] In addition, the observed higher prevalence in our setting may be due to cultural differences in the settings or reluctance of women in the sub-Saharan region to talk openly about sex and seek solutions for their sexual concerns.[11, 22, 23, 25, 47-51] The various explanations for the observed differences have also made generalization and comparison of studies challenging. This study highlights the significant burden of sexual dysfunction within the HIV-positive population in Nigeria, a concern that is particularly acute when considered against the backdrop of the already high prevalence (63%) of sexual dysfunction among women in general in the country [52]. This study used a validated and standardized tool and a specific population group to ensure generalization to our population group and context. This study confirmed that HIV infection is associated with high sexual dysfunction and thus deserves close attention.[22, 23, 25, 47-51]

Line 228: is this referring to global frequency of abnormality or to the Nigerian setting?

Response:

Thank you, and we have revised the information, see line 307-311.

As anticipated, the occurrence of abnormalities in specific sexual function domains is more pronounced in the Nigerian HIV-positive population studied than in the general Nigerian population. However, this frequency aligns with what has been reported in other studies focusing on HIV-positive women, indicating a consistent pattern of sexual dysfunction among HIV-positive women across different geographical and cultural contexts.[25, 48-50, 69]

Line 235: “More than half of the respondents” – review sentence as advised before.

Response:

Thank you, and we have revised the information, see line 316-318.

In this study, a significant majority of respondents experienced dysfunction in at least three specific sexual function domains, with 76.8% reporting issues with desire, 66.0% with arousal, and 50.0% with orgasm.

Line 254: Menopausal status instead of “A menopausal state” maybe better to use.

Response:

Thank you, and we have revised the information, see line 273-275.

Menopausal status was also found as a risk factor for sexual dysfunction in other studies involving women with known HIV status and the general population

Line 256: it is Possible – why not quote a study that prove the link with a hormonal imbalance in menopause with sexual dysfunction? To say it is possible, leave it open for speculation and I think science has established this association long ago.

Response:

Thank you, and we have revised the information, see line 267-275.

The association between changes in blood hormonal levels in women during menopause and a decline in sexual functioning is well-established in the scientific literature. Studies have consistently demonstrated that the hormonal imbalances experienced during menopause, particularly the reduction in estrogen and testosterone levels, are significantly linked to various aspects of sexual dysfunction. In a comprehensive review by Nappi and Lachowsky (2009) highlights the direct correlation between decreased estrogen levels and reduced vaginal lubrication, decreased libido, and other sexual health issues in menopausal women [58]. Menopausal status was also found as a risk factor for sexual dysfunction in other studies involving women with known HIV status and the general population.[11, 59-61]

Line 267 & 271: Ref 33 – Erectile dysfunction is associated with Hypertension and this study is done on women. Although it is a well-known sexual problem associated with hypertension, how relevant is this reference in an article reporting on findings in a female population? Do we have and specific sexual dysfunction associated in females with hypertension in the literature? Maybe refer to this systematic review for further research?

Santana, L.M., Perin, L., Lunelli, R. et al. Sexual Dysfunction in Women with Hypertension: a Systematic Review and Meta-analysis. Curr Hypertens Rep 21, 25 (2019). https://doi.org/10.1007/s11906-019-0925-z

Response:

Thank you, and we have revised the information, see line 285-296.

Hypertension and diabetes are prevalent comorbidities in our environment, particularly among the HIV patient population, [64] and 61-88% of patients with these diseases experiencing sexual dysfunction.[11, 25, 37, 65] While erectile dysfunction is a well-recognized consequences of hypertension in men, women with hypertension also face significant sexual health challenges, including reduced sexual desire, arousal issues, and overall dissatisfaction with sexual life. The conclusive evidence of this association between hypertension and various forms of sexual dysfunction in women is well-documented in a systematic review and meta-analysis by Santana, L.M., Perin, L., Lunelli, R. et al. (2019).[66] Given the high prevalence of sexual dysfunction among patients with these comorbid conditions, it is not surprising that co-morbid medical conditions retained an independent statistical association with sexual dysfunction, even after controlling for other possible confounders. This association is consistent with findings from other studies in our environment.

Line 280: the word potency – what does it mean? Sexual interest and potency? Why not use one of the 6 domain words from the FSFI like desire?

Response:

Thank you, and we have revised the information, see line 333-339.

Castelo-Branco et al.,[71] in their study among healthy women living in Santiago de Chile, found desire disorder (38.0%) the most common, followed by arousal disorder (32.0%), pain disorder (33%), and orgasm disorder (25.0%). In a study by Kadri and colleagues[73] among women living in Morocco, desire disorder was reported by 18%, orgasm disorder by 12%, arousal disorder by 8%, and pain disorder by 8%. The inference that can be made from the above is that while the severity of specific domain dysfunction may be worse in HIV-positive women, their distribution and pattern of occurrence are like those in HIV-positive and HIV-negative women.[11, 22, 23, 25, 61, 73]

Line 284-289: Discussion disclosure of HIV status and the fears surrounding it and referring the violence – please consider reference this statement?

Response:

Thank you, and we have revised the information, see line 326-332.

Castelo-Branco et al.,[71] in their study among healthy women living in Santiago de Chile, found desire disorder (38.0%) the most common, followed by arousal disorder (32.0%), pain disorder (33%), and orgasm disorder (25.0%). In a study by Kadri and colleagues[73] among women living in Morocco, desire disorder was reported by 18%, orgasm disorder by 12%, arousal disorder by 8%, and pain disorder by 8%. The inference that can be made from the above is that while the severity of specific domain dysfunction may be worse in HIV-positive women, their distribution and pattern of occurrence are like those in HIV-positive and HIV-negative women.[11, 22, 23, 25, 61, 73]

The discussion address 5 of the 6 independently associated risk factors identified. There is no discussion about the PHQ score >10. As in the introduction – it is really not making sense why this has been included in the study methodology and mentioned in the results, but not put into perspective in the introduction and not discussed in the discussion? (It should be included, the evaluation of Depression is relevant to FSD). Address the omission in the 2 sections: Introduction and Discussion.

Consider consistency. Discuss the 6 factors in the exact order as before:

• PHQ score >10,

• postmenopausal state,

• co-morbid medical conditions,

• HIV disclosure status,

• use of PI’s.

Keeping the same order makes the reading and understanding of the findings so much easier to read and remember for a reader, especially if you have a non-academic reading the paper.

Response:

Thank you, and we have revised the information, see line 252-263.

The result PHQ-9 score of ≥10 is (cOR) of 2.5 (95% CI: 1.6 – 4.0) shows that sexual dysfunction is associated with depressive symptoms. A study by Atlantis & Sullivan (2012) found a strong link between depression and sexual dysfunction, suggesting that the presence of depressive symptoms can significantly increase the likelihood of experiencing difficulties with sexual function.[53] A study by Clayton et al. (2014) found similar results, noting that depression was a significant predictor of sexual dysfunction in women, and suggested that treatment of depressive symptoms could have beneficial effects on sexual function.[54] The study by Mathew & Weinman (2009) also found a strong association between depression and sexual dysfunction, further supporting the findings of this study.[55] However, while these results are compelling, it's essential to consider them in the context of potential confounders and the complexity of sexual health. Factors such as medication side effects, relationship issues, and other psychological or physical health conditions can also significantly impact sexual function. The relationship between depression and sexual dysfunction is likely to be bidirectional and influenced by a range of other factors.[56, 57]

Conclusion

Line 304 – 305: why include Sub-Saharan Africa in this sentence?

Response:

Thank you, and we have revised the information, see line 355-363.

Sexual dysfunction is a significant public health challenge among HIV-positive women in Nigeria, where HIV is endemic. The sexual dysfunction rate of 71.4% confirms that it is a significant challenge in our setting. Sexual dysfunction in the six domains ranged from 38.8% in the satisfaction domain to 76.8% in the desire domain. Six factors are significantly associated with sexual dysfunction: menopausal state, PHQ score of 10 and above, opportunistic infection, co-morbid medical condition, use of protease-based antiretroviral regimen, and non-disclosure of HIV status. There is a need to sensitize and train healthcare workers to assess for sexual challenges during routine programmatic consultations. The national HIV programs should, as a matter of urgency, include screening and treatment of sexual dysfunctions among HIV-positive women in the country’s treatment guidelines.

---

## [Editor Report · Decision Letter 2]

20 Mar 2024

Sexual Dysfunction among Nigerian Women Living with HIV Infection

PONE-D-23-24987R2

Dear Dr. Akinsolu

We’re pleased to inform you that your manuscript has been judged scientifically suitable for publication and will be formally accepted for publication once it meets all outstanding technical requirements.

Kind regards,

Deidre Pretorius, PhD

Academic Editor

PLOS ONE
---

## [Editor Report · Acceptance letter]

29 Mar 2024

PONE-D-23-24987R2 

PLOS ONE

Dear Dr. Akinsolu, 

I'm pleased to inform you that your manuscript has been deemed suitable for publication in PLOS ONE. Congratulations! Your manuscript is now being handed over to our production team.

Kind regards, 

on behalf of

Dr. Deidre Pretorius 

Academic Editor

PLOS ONE